# Recent history and future demise of Jostedalsbreen, the largest ice cap in mainland Europe

Henning Åkesson<sup>1</sup>, Kamilla Hauknes Sjursen<sup>2</sup>, Thomas Vikhamar Schuler<sup>1</sup>, Thorben Dunse<sup>2</sup>, Liss Marie Andreassen<sup>3</sup>, Mette Kusk Gillespie<sup>4</sup>, Benjamin Aubrey Robson<sup>5</sup>, Thomas Schellenberger<sup>1</sup>, and Jacob Clement Yde<sup>2</sup>

Correspondence: Henning Åkesson (henning.akesson@geo.uio.no)

**Abstract.** Glaciers and ice caps worldwide are in strong decline, and models project this trend to continue with future warming, with strong environmental and socio-economic implications. The Jostedalsbreen ice cap is the largest ice cap on the European mainland (458 km<sup>2</sup> in 2019) and occupies 20% of the total glacier area of mainland Norway. Here we simulate the evolution of Jostedalsbreen since 1960, and its fate in a changing climate in the 21st-century and beyond (2300). This ice cap consists of glacier units with a great diversity in shape, steepness, hypsometry, and flow speed. We employ a coupled model system with higher-order three-dimensional ice dynamics forced by simulated surface mass balance that fully accounts for the mass-balance elevation feedback. We find that Jostedalsbreen may lose 12-74% of its present-day volume until 2100, depending on future greenhouse gas emissions. With mid-range results obtained using the climate model ECEARTH/CCLM, Jostedalsbreen is projected to lose 49% (RCP4.5) and 63% (RCP8.5) of its contemporary ice volume by 2100. Regardless of emission scenario, the ice cap is likely to split into three parts during the second half of the 21st century. Our results suggest that Jostedalsbreen will likely be more resilient than many smaller glaciers and ice caps in Scandinavia. However, we show that by the year 2100, the ice cap may be committed to a complete disappearance during the 22nd century, under high emissions (RCP8.5). Under medium 21st-century emissions (RCP4.5), the ice cap is bound to shrink by 90% until 2300. Further simulations indicate that substantial mass losses undergone until 2100 are irreversible; the ice cap would not recover to its contemporary volume if the future surface mass balance was reversed to that of the present-day. Our study demonstrates a model approach for complex ice masses with numerous outlet glaciers such as ice caps, and how tightly linked future mass loss is to future greenhousegas emissions. Finally, uncertainties in future climate conditions, particularly precipitation, appear to be the largest source of uncertainty in future projections of maritime ice masses like Jostedalsbreen.

<sup>&</sup>lt;sup>1</sup>Department of Geosciences, University of Oslo, Oslo, Norway

<sup>&</sup>lt;sup>2</sup>Department of Civil Engineering and Environmental Sciences, Western Norway University of Applied Sciences, Sogndal, Norway

<sup>&</sup>lt;sup>3</sup>Norwegian Water Resources and Energy Directorate (NVE), Oslo, Norway

<sup>&</sup>lt;sup>4</sup>VIA University College, Nørre Nissum, Denmark

<sup>&</sup>lt;sup>5</sup>Department of Earth Science, University of Bergen, Bergen, Norway

## 1 Introduction

20

50

Melting glaciers and ice caps are powerful and concrete symbols of climate change, and can raise awareness and spur climate action among people who visit them (Dannevig and Rusdal, 2023). Ice caps are dome-shaped ice masses with radial flow, often covering the underlying highland topography. They consist of numerous connected glacier units (e.g. Zemp and Haeberli, 2007), which are referred to as outlet glaciers if their lower reaches are separated by mountain areas. Since the year 2000, glaciers and ice caps have contributed nearly as much to sea-level rise as the combined mass loss from the Greenland and Antarctic ice sheets (Hugonnet et al., 2021). In addition, the response of glaciers and ice caps to global warming has widespread societal implications at regional to local scales. This includes consequences for tourism, hydropower production, agriculture and local ecosystems as a result of changes to glacier extent and the magnitude and timing of meltwater runoff (e.g. Milner et al., 2017; Huss et al., 2017). Such implications are particularly strong at Jostedalsbreen ice cap, the largest ice mass in mainland Norway and Europe, and the focus of this study. Meltwater from this ice cap feeds several hydropower stations, and Jostedalsbreen attracts over 600,000 visitors every year (Jostedalsbreen nasjonalparkstyre, 2021), with three glacier visitor centres, a dedicated museum and several curated glacier viewpoints. The ice cap is extensively used for recreation and exploration, including skiing and glacier hiking. Visitors play an important role for the livelihood of many local settlements in the vicinity of the ice cap.

The primary objective of this study is to assess the short- and long-term fate of Jostedalsbreen under future climate change. To achieve this, we demonstrate a comprehensive modelling approach of a topographically complex ice cap, consisting of 81 connected glacier units that are exceptionally diverse in shape, steepness, hypsometry and flow speed. This setting ranks among the most challenging real-world cases outside the polar ice sheets. The wide range of characteristics of the individual glacier units found at Jostedalsbreen make them representative of glaciers found in many glacierised regions of the world. Modelling the evolution of Jostedalsbreen can be viewed as an application to a small mountain range with many connected glaciers, an intermediary step between glacier-specific studies and large-scale regional or global applications. The treatment of so many connected glaciers introduces a number of challenges, which are less prominent or absent for studies of single glaciers or idealised geometries. This includes the need for detailed historical and contemporary input data, spatially constrained model parameters embedded with advanced ice-flow physics and surface mass balance, and climate projections suitable for a mountainous region. In a warming climate, the positive feedback between ice thinning and stronger surface melt at increasingly lower surface elevations, and associated dynamic adjustments, is also crucial to capture. This mass-balance elevation feedback (Harrison et al., 2001) is particularly important for ice caps (Åkesson et al., 2017), where minor changes in elevation in their flat interior regions affect the surface mass balance over large areas.

Detailed modelling studies of single glaciers or ice caps have improved the physical realism in glacier models, evaluated the importance of physical processes and climate forcing, and projected future evolution and associated impacts on nature and society (e.g. Oerlemans, 1997; Aðalgeirsdóttir et al., 2005; Jouvet et al., 2009; Giesen and Oerlemans, 2010; Ziemen et al., 2016; Ekblom Johansson et al., 2022). Therefore, the rationale for in-depth studies of individual glaciers and ice caps remains strong (Zekollari et al., 2022). Such studies often resolve ice dynamics in two dimensions with simplified physics (e.g.

Le Meur and Vincent, 2003; Giesen and Oerlemans, 2010; Ziemen et al., 2016; Åkesson et al., 2017; Schmidt et al., 2020). This is usually viable for simple glacier geometries, slow-flowing ice, and long time scales (Leysinger Vieli and Gudmundsson, 2004; Le Meur et al., 2004; Adhikari and Marshall, 2012). In contrast, ice caps may consist of numerous connected glaciers with variable geometry and dynamics, controlled to some extent by the underlying topography, often with fast-flowing outlet glaciers. For example, assumptions of shallowness (Hutter, 1983) commonly used in many ice-flow models are not justified for complex ice caps. This dynamic setting may require three-dimensional (3-D) modelling efforts with higher-order physics (Adhikari and Marshall, 2012; Zekollari et al., 2017), which only occurs as exceptions in the literature (e.g. Adhikari and Marshall, 2012; Gilbert et al., 2016; Zekollari et al., 2017).

While ice caps present challenges for detailed studies, these issues are exacerbated in regional or global glacier-evolution models. In such models, interactions between ice dynamics, geometry evolution, and glacier surface mass balance are typically represented in even simpler terms (Huss et al., 2012; Huss and Hock, 2015; Maussion et al., 2019; Rounce et al., 2023), and each glacier unit treated individually, often using simple one-dimensional (1-D) flowline dynamics (Maussion et al., 2019; Rounce et al., 2023). In a warming climate, ice caps may split up into separate glacier units, ice divides may migrate, or the underlying topography can emerge in the middle of a glacier unit due to ice thinning. These future geometric changes are likely to occur in many glacierised areas of the world. Here, the glacier-by-glacier approach will suffer, and the 1-D flowline dynamics will no longer be valid. As such, large-scale glacier evolution models struggle to represent some of the most dynamic ice masses on the planet outside of the polar ice sheets (Millan et al., 2022). More detailed knowledge of the evolution and dynamics of ice caps, and how to best represent these changes in models, is therefore urgently needed (Zekollari et al., 2022). There is also a need to bridge the two end-members of glacier-scale simulations and regional-to-global-scale approaches.

A major challenge when simulating glacier dynamics is how to account for friction between ice and the underlying bedrock. Basal friction strongly influences glacier flow speeds (e.g. Iken, 1981), but is difficult to observe directly. Subglacial bulk properties are therefore usually inferred from surface velocities in the form of a spatially variable basal friction parameter (Morlighem et al., 2010; Gillet-Chaulet et al., 2012). This approach is a cornerstone in ice-sheet projections, supported by remote sensing datasets of ice-sheet wide surface speeds. In contrast, modelling efforts for glaciers and ice caps have not been able to fully take advantage of these spatially continuous velocity datasets. This is because the spatial resolution and/or temporal coverage of remote-sensing products often have been too poor to properly resolve the details of ice flow of glaciers and ice caps. Therefore, glacier modelling studies have often imposed some ad-hoc relationship for the friction parameter (Zekollari et al., 2022). The advent of detailed datasets of velocity for every glacier in the world (Friedl et al., 2021; Millan et al., 2022) presents new opportunities for friction inversions in detailed modelling studies. These datasets can however not be uncritically applied 'off-the-shelf', especially not for complex ice masses such as ice caps. For example, the accuracy of the underlying method may be smaller than the actual ice movement, or the data may represent snapshots or subsets of shorter periods, and hence not representative for annual mean velocities.

Another common challenge for modelling studies is poorly known ice thickness. The vast majority of Earth's over 200,000 glaciers do not have measurements of ice thickness (Welty et al., 2020). This means that model applications must revert to global and regional thickness products (Farinotti et al., 2019; Millan et al., 2022; Frank and van Pelt, 2024), which may not be

85

accurate for detailed applications (Gillespie et al., 2024). Some of these thickness models struggle to represent connected ice complexes, because each glacier unit is treated individually (e.g. Farinotti et al., 2019; Millan et al., 2022). This usually leads to issues with thickness along ice divides.

Poor representation of the subglacial topography may bias modelled glacier dynamics, which makes future glacier projections and associated local impacts uncertain at best.

A key conceptual and practical challenge is that the climate system is inert. Glaciers and ice caps are excellent examples of this. It can take years to decades for a glacier to respond significantly to a change in climate (e.g. Jóhannesson et al., 1989), and centuries to millennia before ice mass loss and retreat is manifested in a new steady-state (e.g. Marshall, 2005). To simulate the historical evolution of a glacier, however, a realistic starting point is essential (Aschwanden et al., 2013). A biased initial model state will strongly affect historical and future model evolution (Ađalgeirsdóttir et al., 2014), yet obtaining this initial state with the correct climatic and dynamic memory imprinted is notoriously difficult. For example, the input data and climate forcing may be lacking, patchy or uncertain, or the model physics may be incomplete. These challenges are hard enough to resolve for a single glacier, and becomes increasingly difficult for the great number of glaciers involved in comprehensive applications like the one presented here.

In terms of future change, climate science has been largely preoccupied with the 21st century, not least reflected and refuelled by the landmark establishment of the Paris agreement in 2015 (Meinshausen et al., 2020), with the goal of limiting global warming to 1.5–2 °C above pre-industrial temperatures. Meeting the Paris goals is, at the time of writing, still technically feasible but extremely challenging, while the stakes are high for both nature and people worldwide (United Nations Environment Programme, 2024). To understand the full implications of current and near-future climate change, we however also need to consider future inertia and longer timescales. The concept of 'committed glacier mass loss' is around in the literature (e.g. Goldberg et al., 2015), sometimes referred to as mass loss 'in the pipeline', and yet deserves to be assessed in greater detail. Committed ice mass loss in a glaciological context refers to what will happen to an ice mass with no further change in climate forcing, that is, the response under a continuation of the recent climate into the future (e.g. Box et al., 2022). For example, a large committed mass loss would imply a glacier in strong disequilibrium with climate. Similarly, zero committed mass loss would mean a glacier which has adapted fully to recent climate forcing, reaching a complete steady-state; something which never occurs in reality and only exist in theory and models. Committed mass loss is a pertinent concept as we are heading towards the post-Paris agreement world beyond year 2100, where carbon levels in the atmosphere and global temperature rise may stabilise, or eventually even be reversed. Some studies consider the fate of glaciers and ice caps beyond the year 2100 (e.g. Schmidt et al., 2020), but these are exceptions. There is certainly a need to consider these time scales in the glaciological context and in climate science at large (Meinshausen et al., 2020). In doing so, local and global policymakers can be informed about expected ice mass loss or glacier retreat, now and in the future.

In this study, we make progress from the modelling of individual glacier units towards detailed modelling studies of entire regions. For Jostedalsbreen, previous studies have focused on the evolution of single outlet glaciers (Nigardsbreen and Briksdalsbreen; Fig. 1) using 1-D flowline models (Oerlemans, 1997; Laumann and Nesje, 2009), or surface mass balance modelling (Jóhannesson et al., 1995; Sjursen et al., 2025). However, no study has previously attempted to simulate the historical or future

evolution of the entire ice cap. We employ a coupled model system with 3-D ice dynamics, and surface mass balance simulated using a temperature-index model with spatially constrained parameters calibrated with a Bayesian approach (Sjursen et al., 2025). A newly acquired detailed dataset on ice thickness and bed topography (Gillespie et al., 2024) allows for realistic 3-D simulations of historical and future changes of the complex ice cap.

First, we simulate historical ice-cap changes since the 1960s. These simulations generate a plausible initial state for the future projections, and is used to validate our model setup against observations of ice thickness, ice margins and ice velocities. We then launch coupled simulations to the year 2100, using two scenarios of future greenhouse-gas emissions. Finally, we assess committed mass loss towards year 2300, the potential for ice-cap recovery after year 2100, as well as the possibility for glacier regrowth from ice-free conditions in the current and future climates.

## 2 Study area

Jostedalsbreen (61° 40'N; 7° 00'E) is a northeast-southwest-oriented ice cap located c. 100 km inland from the west coast of Norway (Fig. 1). It is the largest coherent ice mass in mainland Europe with an area of 458 km² in 2019 (Andreassen et al., 2022), covering an elevation range between 381 and c. 2001 m a.s.l. (Andreassen et al., 2022; Kjøllmoen et al., 2024). The maximum ice thickness is about 630 m, and the ice volume is calculated to  $70.6 \pm 10.2$  km³ (in 2018-2023; Gillespie et al., 2024). The ice cap is divided into 81 glacier units with great diversity in size, shape, steepness, and orientation (Fig. 1b; Andreassen et al., 2022). The larger glacier units are outlet glaciers terminating in glacial valleys. Glacier ice-surface catchments are mainly determined by the bed and surrounding topography, although some discrepancies exist (Gillespie et al., 2024). Ice velocities reach up to a few hundred meters per year for the fastest outlet glaciers Nigardsbreen and Tunsbergdalsbreen, while most outlet glaciers typically move a few tens of meters annually (Wangensteen et al., 2006; Nagy and Andreassen, 2019).

Radar measurements suggest that the ice is predominantly temperate, but with isolated patches of cold ice (Gillespie et al., 2024). The ice cap is situated directly on Proterozoic gneiss bedrock belonging to the Western Gneiss Region of Norway (Hacker et al., 2010), except for the lower part of the outlet glacier Austerdalsbreen, which likely rests on unconsolidated glaciofluvial and till sediments (Seier et al., 2024).

Since the maximum extent of Jostedalsbreen during the Little Ice Age (c. 1740 to 1860 Gjerde et al., 2023), the ice cap has lost about 20% in area and volume (Carrivick et al., 2022). This recession has occurred progressively with decadal-scale interruptions of glacier advances or stillstands, as evidenced by sequences of moraine ridges deposited in front of most outlet glaciers (e.g. Erikstad and Sollid, 1986; Bickerton and Matthews, 1993). The latest advance of Jostedalsbreen's outlet glaciers occurred in the 1990s to early 2000s, and was a response to high winter snowfall in the preceding years (Andreassen et al., 2005; Nesje and Matthews, 2012). This illustrates that large-scale atmospheric circulation patterns, such as the North Atlantic Oscillation, influence the surface mass balance of Jostedalsbreen (Nesje et al., 2000). The current climate is characterized by mean annual air temperatures between -3 and +5°C and precipitation amounts between c. 1000 and c. 2000 mm a<sup>-1</sup>, where most precipitation falls in the western part of the ice cap (Carrivick et al., 2022; Sjursen et al., 2025).

Jostedalsbreen has been extensively studied for more than a century. Long-term records of annual glacier front variation exist for six outlet glaciers (Austerdalsbreen, Brenndalsbreen, Briksdalsbreen [stopped in 2015], Fåbergstølsbreen, Nigardsbreen, Stigaholtbreen) dating back to 1899 (Andreassen and Elvehøy, 2021). Surface mass balance has been monitored for several outlet glaciers, but only the two records at Nigardsbreen (back to 1962) and Austdalsbreen (back to 1988) are maintained today (Andreassen et al., 2020; Kjøllmoen et al., 2024).

## 160 3 Glacier input data





## 3.1 Glacier geometry

To simulate the evolution of Jostedalsbreen, we need historical and contemporary geometries of the ice cap. Ice-surface to-pography of 10 m resolution for the present day (2020 Digital Terrain Model; DTM) is used in calibration of the ice-flow model dynamics, as well as a final target in historical simulations (Sections 5.1 and 5.2). The source of the surface DTM on Jostedalsbreen is a lidar survey from 2020, except for the lower tongue of Tunsbergdalsbreen where the survey year is 2017 (Andreassen et al., 2023). Ice thickness has been extensively measured at Jostedalsbreen over the years 2018–2023 using ground- and helicopter-based ice radar. A distributed gridded dataset of ice thickness has been produced from these point measurements, by applying an ice-thickness model based on the inversion of surface topography (Gillespie et al., 2024). We hereafter refer to this dataset as the *observation-based* ice thickness. To derive the subglacial topography of Jostedalsbreen, this observation-based gridded ice thickness (Gillespie et al., 2024) was subtracted from the present-day ice-surface DTM. To produce a seamless bed topography map of the ice cap and nearby ice-free areas, we used the software QGIS (QGIS Development Team, 2024) to merge the subglacial topography with a present-day 10 m DTM of areas without glacier cover.

Meanwhile, another DTM is needed as a starting point for the historical simulations 1960–2020 outlined in Sect. 5.2. Andreassen et al. (2023) presented a DTM for 1966 based on photogrammetric reconstruction of historical aerial photographs. However, this DTM does not cover southern Jostedalsbreen. To produce a DTM for 1966 covering the entire ice cap, we extended the DTM from Andreassen et al. (2023) using another 10 m DTM produced by the company Hexagon (Gulbrandsen, 2022). This DTM is based on the same 1966 aerial photographs and has a greater spatial extent, but is more noisy. The resulting DTM covered the entire ice cap, but contained data gaps over steep terrain and areas with low image contrast. We therefore georeferenced the N50 1966 topographic map from the Norwegian Mapping Authority (map sheet 1318-2; Paul et al., 2011), manually digitised the 20 m interval contour lines over the data voids, and interpolated the elevation using the TopotoRaster function within ArcGIS Pro (ArcGIS Development Team, 2025). This interpolation tool is based on the Australian National University DEM (ANUDEM) algorithm, designed to produce a hydrologically accurate DTM that preserves ridgelines and stream networks (Hutchinson et al., 2011). These interpolated elevations were then used to fill the voids in the DTM. In the southern part of the ice cap, the resulting DTM had some artefacts at the boundaries between the high-resolution DTM from Andreassen et al. (2023) and the 1966 topographic contour map. These were smoothed using a 5x5 low-pass filter. The remaining traces of these artefacts quickly dissipated during the spinup simulation of the ice cap (Sect. 5.1). Between 1966 and 2020, there is no data of ice-surface topography available that covers the entire ice cap.

**Figure 1.** Ice-surface topography of Jostedalsbreen ice cap. Key outlet glaciers are named. Observed ice-cap extents in 1966 and 2019 are shown (Andreassen et al., 2023), as well as individual glacier catchments (blue). The inset map shows the location of Jostedalsbreen in southern Norway. Coordinates in UTM 33N, datum ETRS89 on main map and in geographical coordinates on inset.

## 3.2 Ice velocity






To construct a representative present-day velocity dataset for calibration and validation of the ice-flow model (Sect. 5.1), we use a global velocity product as a baseline (Millan et al., 2022). Upon close inspection of this product, which is based on optical and synthetic aperture radar (SAR) imagery from 2017–2018, we discovered several artifacts. Mainly, we found spurious velocity magnitudes for flat areas high on the ice-cap plateau where image contrast or amplitude patterns are typically low. For example, ice along the ice divide of Nigardsbreen supposedly flows more than 100 m a<sup>-1</sup>, while in reality, surface velocities near ice divides are typically limited to a few meters per year. To mediate unrealistic velocities close to ice divides, velocity maps were derived using InSAR from European Remote Sensing satellite (ERS) from 1996. Velocity maps with partial coverage of Jostedalsbreen were acquired by differential InSAR performed on three ERS-1/ERS-2 tandem pairs. The ascending pairs were acquired on 20/21 January 1996 and 11/12 February 1996, the descending on 22/23 March 1996. From those pairs, we derived velocities in the look direction of the satellite ("line-of-sight", incidence angle = 23.2 degrees), which were then combined to a two-dimensional, line-of-sight velocity map. The differential InSAR processing was performed using the GAMMA Remote Sensing software (GAMMA AG, 2016). The 1996-velocity map was used to correct the global data from Millan et al. (2022) in areas with surface slopes lower than 0.08 and/or above 1775 m a.s.l. Judging from velocity data from complementary sources (Table A1), ice flow at the ice divides displays little year-to-year variability. Moreover, elevation changes at the ice-cap plateau since 1966 have been very small (less than  $\pm 5$  m; Andreassen et al., 2023), which suggests a negligible velocity change around the ice divides since 1996. We therefore judge our composite velocity dataset to be robustly constructed.

Overall, the global dataset from Millan et al. (2022) appears to be more representative for fast-flowing regions at lower elevations of Jostedalsbreen, where it is in close agreement with other velocity datasets that cover parts of the same time period (e.g. Nagy and Andreassen, 2019, see Table A1). Velocities from the global dataset are c. 100 to 500 m a<sup>-1</sup> for fast-flowing outlet glaciers, while differences among various data sources are on the order of 50–100 m a<sup>-1</sup> for a given glacier. However, this is not a straightforward comparison. Visual inspection of other velocity products covering parts of the 2010s (Table A1) suggests that velocities in the global dataset may be too high also in some fast-flowing areas. This is possibly because the 2017–2018 global data are biased towards high summer velocities and thus do not represent annual averages. The effect of this bias will vary from glacier to glacier, depending on the magnitude of seasonal velocity variations.

#### 4 Model description

## 4.1 Ice-flow model

et al., 2012) considering 3-D higher order physics (Blatter, 1995; Pattyn, 2003). The model domain comprises the 1966 ice-cap extent and a c. 5 km buffer zone. This ensures an accurate representation of surface mass balance (SMB) around the margins (Sect. 4.3), allows for frontal fluctuations of outlet glaciers, and resolves interactions with nearby glaciers separate from the main ice cap. The finite-element model mesh consists of c. 87,000 horizontal elements spread over four vertical layers. To

capture ice flow in steep terrain, we vary the mesh resolution from 100 to 300 m based on gradients in bedrock topography. In addition, we enforce a high resolution (100 m) for several key catchments (Fig. A1). This improves the representation of the two largest and fastest outlet glaciers Nigardsbreen and Tunsbergdalsbreen (area 41.7 and 46.2 km², respectively; Andreassen et al., 2022), with ice velocities up to 300–500 m/a, and several other glaciers of particular interest to local hydropower and tourism (Austerdalsbreen and glaciers draining into Oldedalen; see Fig. 1). A time step of 0.02–0.05 years (7.3–18.25 days) is used depending on the experiment (Sect. 5), which ensures numerical stability.

Two of the 81 glacier units are lake-terminating glaciers (Austdalsbreen and Sygneskarsbreen in the north), which account for about 3% of the total ice-cap volume. Frontal ablation is not included in the model, and the lake surface is considered the bedrock topography. Neglecting iceberg calving and subaqueous melt may lead to underestimation of glacier mass loss as long as these glaciers remain lake-terminating. The uncertainty in bedrock elevation may affect potential glacier advances, however glacier retreat dominates in our simulations.

Jostedalsbreen is considered a temperate ice cap (Sect. 2) and we assume a uniform ice temperature of 0 °C to compute the associated ice viscosity using an Arrhenius law (Cuffey and Paterson, 2010, p. 75). Glen's flow law is used to compute the strain rate in response to stress at any point in the glacier (Cuffey and Paterson, 2010, p. 55).

## 4.2 Basal friction



We use a linear viscous friction law (Budd et al., 1979) to compute basal drag  $\tau_h$  as

$$\boldsymbol{\tau}_b = -\alpha^2 N \mathbf{u}_b,\tag{1}$$

where  $\alpha$  is a friction parameter,  $\mathbf{u}_b$  is basal velocity and  $N = \rho_i g H$ , where  $\rho_i$ , g, H are ice density, gravitational acceleration and ice thickness, respectively. The friction parameter is allowed to vary from 10 to 400 s<sup>(1/2)</sup> m<sup>(-1/2)</sup>. We do not account for input of surface melt and associated seasonal variations of N and  $\mathbf{u}_b$ .

We constrain the local values of the basal friction parameter  $\alpha$  by inversion using an adjoint method that minimizes the misfit between the modelled u and observed ice velocities  $u^{obs}$ , by means of the following cost function (e.g. Morlighem et al., 2010):

$$\mathcal{J}(\mathbf{u},\alpha) = \gamma_1 \frac{1}{2} \int_{S} \left( \left( u_x - u_x^{obs} \right)^2 + \left( u_y - u_y^{obs} \right)^2 \right) dS + \gamma_2 \int_{S} \left( log \left( \frac{||\mathbf{u}|| + \epsilon}{||\mathbf{u}^{obs}|| + \epsilon} \right) \right)^2 dS + \gamma_3 \frac{1}{2} \int_{B} ||\nabla \alpha||^2 dB$$
 (2)

The first and second terms are the absolute and logarithmic velocity misfits, respectively. The third term is a regularization term to prevent singularities and penalises inferred parameters to avoid overfitting, and  $\gamma_1 = 2000$ ,  $\gamma_2 = 1$ , and  $\gamma_3 = 3.2 \times 10^{-6}$  are respective weights for each of the terms in the cost function Eq. 2. S and B refer to the surface and bed, respectively.

Model velocities in Eq. 2 are computed using a fixed-geometry stress-balance calculation with the present-day data described in Sect. 3 (experiment 'Calibration' in Table 2). We use this inversion method to infer the spatially variable friction parameter  $\alpha$  for the two largest and fastest outlet glaciers Tunsbergsdalsbreen and Nigardsbreen, where we judge our mosaic velocity

**Table 1.** Constants and parameter values used in this study.

| Parameter                  | Symbol     | Value                 | Unit                               |
|----------------------------|------------|-----------------------|------------------------------------|
| Ice density                | $ ho_i$    | 917                   | ${\rm kg}~{\rm m}^{-3}$            |
| Gravitational acceleration | g          | 9.81                  | ${\rm m}~{\rm s}^{-2}$             |
| Rate factor                | A          | $2.4 \times 10^{-24}$ | $\mathrm{s}^{-1}~\mathrm{Pa}^{-3}$ |
| Basal friction parameter   | $\alpha$   | 10-400                | $s^{(1/2)}\;m^{(-1/2)}$            |
| Glen's law exponent        | n          | 3                     |                                    |
| Mesh resolution            | $\Delta x$ | 100-300               | m                                  |
| Time step                  | $\Delta t$ | 0.02-0.05             | a                                  |

dataset described above to be the most representative. For the rest of the ice cap and for ice-free areas, we use a spatially variable friction parameter proportional to bedrock altitude  $z_b$ , following Åkesson et al. (2018):

$$\alpha = \beta_{max} \times \frac{\min[\max(0, z_b + z'), z_b]}{\max(z_b)},\tag{3}$$

where  $\beta_{max} = 400 \ s^{(1/2)} m^{(-1/2)}$  and  $z' = 500 \ m$  a.s.l. This simple parametrisation imposes a higher basal friction at higher elevations and more slippery conditions downglacier.

## 4.3 Surface mass balance model




To simulate SMB, we use the model of Sjursen et al. (2023, 2025) where melt is modelled using a temperature-index approach and accumulation is the sum of solid precipitation, assuming a linear transition between solid and liquid precipitation around a temperature threshold. The model uses 1 km gridded daily mean temperature and daily total precipitation from seNorge\_2018 (Lussana et al., 2019; Lussana, 2021), which is largely based on interpolation of measurements from a large network of weather stations across Norway (see Sjursen et al. (2023) for details). The SMB model has been calibrated to Jostedalsbreen with a Bayesian approach that uses seasonal glaciological observations to constrain accumulation and melt over the period 1962–2020, and decadal satellite-derived geodetic observations for 2000–2019 from Hugonnet et al. (2021) to derive spatially-distributed bias-corrections of temperature and precipitation for each glacier unit (Sjursen et al., 2025). The SMB model domain covers Jostedalsbreen and a 5 km buffer zone around the current ice-cap margins. This makes SMB available as forcing in case of glacier advance, which occurred for some outlet glaciers in the 1990s and early 2000s. We employ SMB simulated using median values of the posterior distributions of model parameters from Sjursen et al. (2025). For consistency, we propagate the glacier-specific temperature and precipitation bias-corrections outside of the glacier margins to fill the remainder of the SMB-model domain.

## 5 Experiment setup








## 5.1 Calibration and spinup of ice-flow model

We perform a dual ice-flow model optimisation to (i) obtain an optimal representation of ice velocities ('Calibration'; Table 2), and (ii) obtain a realistic initial topography for the historical simulation ('Spinup'; Table 2). The success of (i) is measured by the root mean square error (RMSE) between the modelled and observed present-day velocities in a fixed-geometry stress balance simulation. The performance for objective (ii) essentially depends on the accuracy of the entire suite of model choices, assumptions and input data, as detailed in Sect. 3 and 4 (see also Sect. 7.3).

To minimise both (i) the RMSE between modelled and observed present-day velocity during the calibration, and (ii) the RMSE between modelled and observation-based thickness at end-of-spinup, we varied the allowed maximum for the basal friction parameter  $\beta_{max}$ =[200,500] s<sup>(1/2)</sup> $m^{(-1/2)}$  in the inversion. This was done for the large outlet glaciers Nigardsbreen and Tunsbergdalsbreen. Meanwhile, for areas using the friction parameterisation in Eq. 3, we tested different combinations of  $\beta_{max} = [200, 500] s^{(1/2)} m^{(-1/2)}$  and z' = [200, 800] m a.s.l. This process was done iteratively until the lowest possible RMSE for velocity (i) and thickness (ii) was found.

To avoid unrealistic model drift in the historical simulation, we assume that the ice cap was in steady-state in the 1960s. This assumption may be most accurate for steep and short glaciers with fast response times, which mean they are more likely to be in tune with the ambient climate (e.g. Jóhannesson et al., 1989). In contrast, larger outlet glaciers may have been adjusting to a long-term climate signal and hence was not in equilibrium with 1960 conditions. This may introduce some localised bias and influence the historical simulation, as discussed in Sect. 7.3.

Once the optimal model parameters are obtained, we perform a 250-year spinup with a fixed SMB forcing, using the mean annual SMB over the period 1960–1989 (Table 2). During this period, at least six of the glaciers with front-position observations showed little change in terminus positions (NVE, 2025), suggesting that they were in an approximate steady-state. Meanwhile, three glaciers had considerable frontal changes during this period, while five have too little data to determine how close they were to a steady-state. The SMB during this period is therefore considered suitable as forcing for spinup of the 1960s ice cap. We initialise the spinup using the 1966 ice-surface DTM (Sect. 3), and let the ice cap evolve freely until a steady-state is reached, which takes c. 250 years. The 250-year spinup prevents initial model drift when the historical simulation is launched. To measure the performance of the spinup, we calculate RMSE between simulated thickness at the end of spinup, and observation-based thickness in 1966. The latter was derived by subtracting bedrock elevation from the 1966 DTM (Sect. 3.1).

## 5.2 Historical ice-cap evolution

After the calibration and spinup, we launch historical simulations of Jostedalsbreen for the period 1960–2020 ('Historical' in Table 2). This period was chosen because a high-resolution climate reanalysis is available [Sect.4.3;][](Lussana et al., 2019). The 60-year period is also long enough for SMB to have an effect on ice-cap dynamics and outlet glacier frontal variations. It also ensures that recent SMB forcing is imprinted 'in the pipeline' for ice-cap evolution in future simulations (Sect. 5.3.3).

**Table 2.** Overview of experiments performed in this study. Time period refers to year AD for which the experiment is run for, except for Spinup and Regrow-experiments, where no specific time period is modelled. The climate data 'seNorge' refers to the version seNorge\_2018 (Lussana et al., 2019). ECEARTH/CCLM, ECEARTH/HIRHAM, CNRM/CCLM and MPI/CCLM refer to downscaled and bias corrected temperature and precipitation projections (Wong et al., 2016) based on GCM/RCM simulations from EURO-CORDEX (Jacob et al., 2014). SMB is given with years in brackets [yearX-yearY], experiments are performed using the mean annual SMB over the given period. Initial states refer to either the ice-surface data (Sect. 3.1) or modelled ice geometry at the final year of the stated simulations. Note that the 'Calibration' experiment is a fixed-geometry simulation and thus does not involve any climate data or SMB (Sect. 5.1).

| Experiment    | Time period | Climate data   | SMB                   | Initial state |
|---------------|-------------|----------------|-----------------------|---------------|
| Calibration   | 2019        | -              | -                     | DTM2019       |
| Spinup        | 250 yrs     | seNorge        | seNorge[1960-1989]    | DTM1966       |
| Historical    | 1960–2020   | seNorge        | Sjursen et al. (2025) | Spinup        |
| Future4.5-ECC | 2021–2100   | ECEARTH/CCLM   | RCP4.5                | Historical    |
| Future4.5-ECH | 2021-2100   | ECEARTH/HIRHAM | RCP4.5                | Historical    |
| Future4.5-CNC | 2021-2100   | CNRM/CCLM      | RCP4.5                | Historical    |
| Future4.5-MPC | 2021-2100   | MPI/CCLM       | RCP4.5                | Historical    |
| Future8.5-ECC | 2021-2100   | ECEARTH/CCLM   | RCP8.5                | Historical    |
| Future8.5-ECH | 2021-2100   | ECEARTH/HIRHAM | RCP8.5                | Historical    |
| Future8.5-CNC | 2021-2100   | CNRM/CCLM      | RCP8.5                | Historical    |
| Future8.5-MPC | 2021-2100   | MPI/CCLM       | RCP8.5                | Historical    |
| CommitNow     | 2021-2300   | seNorge        | seNorge[2001-2020]    | Historical    |
| Commit4.5     | 2101-2300   | ECEARTH/CCLM   | RCP4.5[2081-2100]     | Future4.5-ECC |
| Commit8.5     | 2101-2300   | ECEARTH/CCLM   | RCP8.5[2081-2100]     | Future8.5-ECC |
| Reverse4.5    | 2101-2300   | seNorge        | seNorge[2001-2020]    | Future4.5-ECC |
| Reverse8.5    | 2101-2300   | seNorge        | seNorge[2001-2020]    | Future8.5-ECC |
| RegrowNow     | 1000 yrs    | seNorge        | seNorge[2001-2020]    | no ice        |
| Regrow4.5     | 1000 yrs    | ECEARTH/CCLM   | RCP4.5[2081-2100]     | no ice        |
| Regrow8.5     | 1000 yrs    | ECEARTH/CCLM   | RCP8.5[2081-2100]     | no ice        |

For historical SMB simulations (1960–2020), we employ daily mean temperature and daily total precipitation from the seNorge\_2018 reanalysis dataset. Meanwhile, we force the ice-flow model with annual SMB averaged from the simulated monthly SMB, since we are not interested in seasonal variations in historical simulations. Regardless, the mesh resolution of 100–300 m is too coarse to capture seasonal frontal variations.


For historical simulations, we forced the ice-flow model with simulated historical SMB (Sect. 4.3) in a one-way fashion (cf. Sect. 5.3.1). The historical runs were initiated with SMB downscaled to the 100 m 1966 DTM and then interpolated onto the mesh of the ice-flow model. We tested two-way coupled historical simulations at an early stage, but the results were nearly

identical to the one-way coupled simulations, since elevation changes 1960–2020 are minor (Andreassen et al., 2023). We therefore continued with the one-way coupling for efficiency.

We use SMB model output with precipitation correction factors adjusted to each individual glacier, as outlined in Sect. 4.3. For both the spinup and historical simulations, we apply a positive SMB correction of 0.75 m w.e. in the northern part of the ice cap (cf. Fig. 1). Without this manual correction, the ice cap thins heavily during spinup, which is not realistic. Modelled SMB suggests that the northern part of the ice cap is the area with the largest mass loss over the spinup period 1960–1989 (Sjursen et al., 2025). In particular, the 1960s display considerable mass loss due to a strong negative anomaly in precipitation amounts. This could be a result of bias in the climate data and/or in the estimated precipitation correction factors for this region. Meanwhile, the assumption of a steady-state ice cap 1960–1989 is possibly less accurate in this area than others. In fact, geodetic mass balance reveals the largest thinning in the northern part of the ice cap over the 1960–2020 period (Andreassen et al., 2023),

The modelled ice-cap state at the end of the historical simulation is validated against present-day surface topography from 2020, and observed ice margins in year 2019. We also compare the model's ability to reproduce present-day ice velocities (Sect. 3.2). A good model performance entails low RMSEs for both ice thickness and velocities, on the order of 20 m and 20 m a<sup>-1</sup>, respectively. Note that the observed ice velocities have already been used in the calibration of basal friction parameters (Sect. 5.1), and should therefore not be viewed as a completely independent dataset for validation.

# 5.3 Future change





## 5.3.1 Coupling of SMB and ice-flow model

**SMB-elevation feedback.** In future simulations, the models for ice dynamics (Sect. 4.1) and SMB (Sect. 4.3) are two-way coupled at yearly intervals. Thereby, we account for the feedback between SMB and the changing glacier-surface elevation over time. At the end of each year, the modelled ice-surface topography is used as input for the downscaling routine described below. SMB is downscaled to the updated ice-surface topography on a monthly basis for the subsequent year, in 100 m resolution. The obtained 100 m SMB is then interpolated onto the mesh of the ice-flow model, which then simulates ice-cap evolution over the following year.

We distinguish between different representations of SMB, to isolate the effects and feedbacks with SMB involved. The reference-surface mass balance, SMB<sub>static</sub>, refers to the SMB on an unchanging reference geometry (no elevation or area changes) and serves as a direct indicator of climatic variations (Elsberg et al., 2001; Huss et al., 2012). The conventional surface mass balance, SMB<sub>transGeom</sub>, is produced with the coupling procedure described above and accounts for the dynamic adjustment of both ice-surface elevation and glacier area to the SMB forcing. To isolate the SMB-elevation feedback from the effect of glacier area changes, we also evaluate SMB<sub>transArea</sub>, which is SMB for an evolving area, but without elevation changes.

**High-resolution downscaling of SMB.** To provide high-resolution distributed SMB to the ice-flow model we downscale the modelled SMB from the 1 km DTM of the seNorge\_2018 dataset to 100 m resolution using the elevation-dependent downscaling procedure of Noël et al. (2016). This algorithm downscales SMB from a coarse to a high-resolution DTM grid

based on linear regression to determine local SMB gradients. We apply the algorithm on a monthly time scale to retrieve monthly fields of SMB at 100 m resolution. First, for each grid cell of the 1 km DTM the local SMB gradient (slope of the regression) is computed using the 1 km SMB and elevation of the given cell and the adjacent cells. The intercept of the regression in the given cell is then found using the 1 km SMB and elevation and the local SMB gradient. Secondly, the 1 km regression coefficients (slope and intercept) are bilinearly interpolated to the 100 m DTM to retrieve 100 m regression coefficients. Finally, SMB is computed in each 100 m grid cell using the 100 m regression coefficients and elevation in the given cell.

## 5.3.2 Future SMB





To simulate SMB over the period 2021–2100, we construct time series of daily mean temperature and precipitation using a combination of daily variability in historical data (seNorge\_2018) and future trends from regional and global climate model (RCM/GCM) projections (Table 2). The general procedure is based on the approach of van Pelt et al. (2021) and can be outlined as follows: (1) we detrend daily temperature and precipitation in seNorge\_2018 for a 20-year historical reference period (2001–2020) by subtracting monthly linear trends in daily temperature and precipitation over the reference period from the historical data, (2) for each combination of climate model and future emission scenario (eight in total) we compute monthly linear trends for 20-year periods (2021–2040, 2041–2060, 2061–2080 and 2081–2100), and establish piece-wise linear functions for future changes in temperature and precipitation, and (3) we superimpose the piece-wise linear functions on the detrended reference data, repeated for each future 20-year future period.

The basis for future trends in temperature and precipitation are 1 km downscaled and bias-corrected daily resolution temperature and precipitation fields (Wong et al., 2016) based on four EURO-CORDEX GCM/RCM combinations (Jacob et al., 2014) and two Representative Concentration Pathway (RCP) emission scenarios that have been used to assess climate change impacts in Norway (Hanssen-Bauer et al., 2017, see Table 2). The original 12.5 km resolution GCM/RCM fields have been re-gridded to 1 km resolution and bias-corrected (see Wong et al. (2016) for details) using 1 km resolution temperature and precipitation fields from the seNorge dataset (version 1.1; Mohr, 2008).

When detrending precipitation, the monthly trends and intercepts are distributed evenly over each day of the month. This results in some days having negative precipitation values, even after the addition of future monthly trends. Where this occurs, daily precipitation is set to zero. This means that the number of wet/dry days in future periods is not necessarily consistent with the reference period. We judge this to be of minor importance since repeating past variability may not be fully representative of future conditions. Capping negative values at zero may also result in deviations between future precipitation sums and computed trends. However, since precipitation generally increases in future projections, large differences are unlikely to occur.

## 5.3.3 Future ice-cap evolution

After ensuring that the model can satisfactorily reproduce recent glacier changes and dynamics, we conduct a suite of experiments of future evolution of the ice cap (Table 2). These simulations are grouped into four categories: 'Future', 'Commit', 'Reverse' and 'Regrow'. Below we describe each group of experiments in more detail.

'Future' simulations (Sect. 6.2.2) are run from 2021–2100, using SMB based on climate forcing from the four GCM/RCM combinations, and for RCP4.5 and RCP8.5 emission scenarios. Each of these simulations, eight in total, start from the transient model state at the end of the 'Historical' simulation. This ensures consistent internal model dynamics and, crucially, that future simulations account for the time-lagged ice-cap response to recent climate forcing. We specifically designed an experiment called 'CommitNow' to assess this aspect, where we let the historical simulation continue until 2300, using the mean annual SMB of 2001–2020 as a constant forcing.

The modelled ice cap in 2100 produced using the ECEARTH/CCLM model combination is used as the initial state for the 'Commit'- and 'Reverse'-experiments (Sections 6.2.3; 6.2.4). This model combination was chosen because it renders a mid-range volume evolution across the four climate model combinations (Sect. 6.2.2). The Commit experiments quantify mass losses 'in the pipeline' by 2100, without any further climate change taking place. They thus shed light on the long-term response of the ice cap to climate change. In these simulations, the ice cap evolves for another 200 years until 2300, using mean annual SMB from 2081–2100 as a constant forcing. Meanwhile, Reverse experiments explore the potential to undo the modelled 21st-century evolution of the ice cap (Reverse4.5 and 8.5). To this end, mean annual SMB of 2001–2020 is used as forcing from 2101 until 2300.

Finally, we assess whether Jostedalsbreen ice cap can regrow from ice-free ice conditions in the present or future climates (Sect. 7.2). These experiments run for 1000 years and use mean annual SMB for the present-day (2001–2020, RegrowNow) or end-of-the-century climate (2081–2100, Regrow4.5 and 8.5). SMB and ice dynamics are coupled every 20 years in these simulations, as opposed to every year in the other future runs. The longer coupling interval for 'Regrow' experiments saves computation time and has here negligible effect compared to coupling every year, due to the relatively slow dynamics involved when growing Jostedalsbreen from no ice.

## 395 6 Results




In the following, we first show the results of our historical (Sect. 6.1) simulation. These provide insight in their own right; no study has previously simulated the recent evolution of Jostedalsbreen as a whole, including ice dynamics. These experiments also (1) generate the initial geometry and volume for future simulations and (2) provide benchmarks that give confidence in the modelled ice dynamics and SMB, and their ability to collectively produce an ice cap that resembles reality. These conditions need to be fulfilled before embarking on future projections (Sect. 6.2).

## 6.1 Historical ice-cap change 1960–2020

Our simulations show that the modelled ice volume of Jostedalsbreen ice cap has changed little since the 1960s. The simulated present-day volume is in excellent agreement with the observed (Fig. 2a). Meanwhile, the modelled volume in 1966 is a few km<sup>3</sup> lower than observed. This underestimation may partly be a result of the practical assumption that the ice cap was in steady-state in the 1960s (Sects. 5.1 and 7.3). On the other hand, the uncertainty in observed ice volume for both the 1960s and

present-day is around 10 km<sup>3</sup>, so the mismatch is well within the uncertainty of the underlying observational datasets of bed topography and ice-surface topography (Sect. 3.1).

Partitioning the volume change 1960–2020 reveals that modelled and observed volumes agree very well for the South, Central and North parts of the ice cap (Fig. 2a; cf. Fig. 1). Of these, the central ice cap amounts to c. two-thirds of the total ice volume of  $70.6\pm10.2~\rm km^3$ . The modelled SMB and ice-volume evolution can be divided into three distinct periods during the historical period: approximate balance 1960–1989, apart from slightly negative SMB in the early 1960s; positive SMB and ice-cap growth in the 1990s; and mass loss after the year 2000 (Fig. 2ac). The reader is referred to Sjursen et al. (2025) for further details on the historical SMB.

The overall agreement between modelled and observed ice margins is good (Fig. 3), especially considering the complex topography and great diversity in glacier characteristics. This agreement includes the large outlets Austerdalsbreen, Tunsbergdalsbreen and Nigardsbreen on the eastern side of the ice cap (cf. Fig. 1). However, the model overestimates the total area by 23% (modelled area 566 km² in 2020; Fig. 2b), mainly due to too advanced margins in the north, where some areas with relatively coarse model elements contribute to a too large modelled areal extent. These areas generally host very thin ice (less than 50 m; Fig. 3a), which means that they contribute little to total ice volume (Fig. 2a). The modelled area shown in Fig. 2be includes all areas with a modelled ice thickness of 10 m or more. If instead all areas with more than 20 m thick ice is considered as glacierised, the modelled area of 2020 becomes 520 km², which reduces the overestimation of areal extent from 23% to 13% (Fig. A2). This means that the present-day modelled ice cap hosts large areas (46 km²) of ice only 10–20 m thick, some which are even disconnected with the main ice cap. This sensitivity analysis, together with a visual inspection of the modelled ice-cap margins in Fig. 3b, suggests that the model performs well in representing both ice volume and area.






Overall, the modelled thickness agrees well with observation-based gridded thickness (RMSE = 24 m; Fig. 3b). Tunsbergdalsbreen, the longest (28 km) and thickest (maximum 626 m; Gillespie et al., 2024) glacier in Norway, sees excellent agreement for ice thickness (RMSE c.  $\pm 30$  m), with deviations within the uncertainty of the observation-based ice-thickness data for this glacier (c. 30–80 m for Tunsbergdalsbreen; Gillespie et al., 2024). This glacier alone contains roughly as much ice (10.8 km³) as the entire South (12.3 km³) and North (8.5 km³) parts of the ice cap, respectively (Gillespie et al., 2024). For the lower part of Nigardsbreen, the present-day ice surface and glacier front are reproduced within c.  $\pm 15$  m, while the upper part is somewhat thinner than the observation-based thickness, with thickness misfit similar to the upper limit of uncertainty for the observation-based dataset (c. 20–50 m for the upper part of Nigardsbreen, Fig. 3b; Gillespie et al., 2024).

Conversely, the model overestimates ice thickness by c. 20–100 m for a few very steep, narrow outlet glaciers, for example Briksdalsbreen and lower Austdalsbreen (Fig. 3b; cf. Fig. 1). For some of these outlet glaciers (e.g. Briksdalsbreen, Brenndalsbreen, Lodalsbreen), this is associated with modelled frontal positions c. 0.5–1 km further downvalley of the present-day observed termini (Fig. 3a). On the other hand, there are also some steep and/or narrow glaciers that are very well represented both in terms of terminus position, thickness and flow speeds (e.g. Fåbergstølsbreen, and several glacier units in the South; Fig. 3)

Velocities are overall well reproduced (RMSE =  $23 \text{ m a}^{-1}$ ), with some spatial variability. For example, it appears that several modelled outlet glaciers are too slow in their steepest regions (e.g. ice falls of Nigardsbreen and Tunsbergsdalsbreen).

**Figure 2.** Simulated historical change (a–c) and future (d–f) evolution with experiment Future 4.5-ECC (cf. Table 2). Recent historical periods of little change (gray), ice-cap growth (blue) and decay (red) are shaded in (a–c). Observed volumes and areas are shown as filled circles in (a) and (b). In (c), historical SMB with evolving model area is shown (SMB<sub>transArea</sub>), but without elevation changes, since these make little difference (Sect. 5.2). Shown in (f) is SMB with transiently evolving geometry (SMB<sub>transGeom</sub>), including the SMB–elevation feedback and area changes.

Figure 3. Comparison between model simulations in this study, and observation-based ice thickness ('obs.') and observed velocity at the end of the historical simulation (2020). (a) Modelled ice thickness; (b) thickness misfit, model - observation-based (Gillespie et al., 2024); (c) modelled annual glacier velocity in 2020 (colours are in log-scale); (d) velocity misfit, model - observed (corrected dataset from Millan et al., 2022). Glacier outlines and divides from 2019 are shown with thin black lines (Andreassen et al., 2022). Ice-free areas within the model domain are shown in grayscale in (a) and (c).

This is true when comparing modelled velocities against the global velocity product (Millan et al., 2022). As mentioned in Sect. 3.2, velocities in this global dataset may be too high in some locations, which would mean that the modelled velocities in fact are more accurate than they appear. Ice thickness and ice margins are indeed well represented in many areas despite too slow apparent velocities. This indicates that the combined SMB–dynamics model system performs well regardless of the discrepancies between modelled and observed velocities.

#### 6.2 Modelled future evolution 2021–2300

## **6.2.1** Reference surface mass balance




Future SMB simulations reveal that substantial changes in SMB forcing (SMB<sub>static</sub>) on Jostedalsbreen can be expected over the coming century (Figs. 4 and 5). Moreover, differences between climate model combinations and emission scenarios are striking (Fig. 4c). The majority of climate models result in cumulative mass losses of around -25 m w.e. over the next 30–40 years, with relatively low variation between the RCP4.5 and 8.5 scenarios. After around year 2050, models show a larger spread with cumulative mass losses up to around -150 m w.e. by the end of the century. Note that this mass loss is for an SMB over a fixed geometry (SMB<sub>static</sub>), that is, before considering the effects of future area changes and the SMB-elevation feedback (see Sect. 6.2.2 and Fig. 11). One model combination stands out from the rest: MPI/CCLM shows a large mass increase under the high-emission scenario (Fig. 4c). This is a result of rapid and extreme (unrealistic; Sect. 7.1) gains in future winter precipitation, along with mainly negative winter temperature and little change in summer temperature (Fig. A5), which renders a positive cumulative SMB<sub>static</sub> over the course of the 21st-century (Fig. 4). The near balance with ECEARTH/HIRHAM under a medium-emission scenario is a result of similar, but less extreme, mechanisms.

Our simulations indicate an increase in the frequency and magnitude of negative SMB throughout the century, under both medium- and high-emission scenarios (Fig. 4a and b, respectively). Under high-emission scenarios, ice-cap wide mass losses can reach SMB magnitudes similar to those seen only for the low-lying tongues of Jostedalsbreen today (e.g. around -6 to -7.5 m w.e. on the lower tongue of Nigardsbreen in 2023; Kjøllmoen et al., 2024). Our results also indicate that, under both emission scenarios, years with positive ice-cap wide SMB can occur far into the 21st century, although this occurs significantly less often during the second half of the century. Uncertainties in future SMB are greater for high-emission scenarios, mainly due to the substantial spread between climate model projections (see Sect. 7.1).

Considering the mid-range climate forcing ECEARTH/CCLM, before considering the SMB-elevation feedback (Fig. A4), the ice cap will retain a substantial accumulation area at high elevations in the South and Central parts for RCP4.5 by the end of the 21st century (Fig. 5a), while annual SMB rates in the North will be mainly negative. For RCP8.5, SMB rates remain positive only for smaller parts on the central plateau (Fig. 5c), with a large reduction in SMB rates over the entire ice cap compared to present day (Fig. 5d). The greatest reduction in SMB occurs on low-lying glacier tongues and along ice-cap margins (Fig. 5b and d). However, the South part, which has shown mostly positive SMB rates over the historical period (1960–2020; Sjursen et al., 2025), also displays large negative changes in SMB by the end of the 21st century for this scenario.

**Figure 4.** Modelled glacier-wide annual reference surface mass balance (SMB<sub>static</sub>, fixed DTM 2020 geometry and 2019 area) for Jostedals-breen over the period 2021–2100. Results are presented as median annual mass balance across climate model combinations for (a) RCP4.5 and (b) RCP8.5, and (c) cumulative mass balance for each model and RCP combination. Bars in (a) and (b) show median modelled mass balance and whiskers show spread of models (minimum and maximum values).

**Figure 5.** Modelled annual reference surface mass balance rates (SMB<sub>static</sub>, fixed DTM 2020 geometry and 2019 area) for Jostedalsbreen by the end of the 21st century (2081–2100 mean) for (a) RCP4.5 and (b) RCP8.5, with ECEARTH/CCLM climate forcing. Difference between end-of-century and present-day (2001–2020 mean) SMB<sub>static</sub> rates for (c) RCP4.5 and (d) RCP8.5.

# 6.2.2 Ice-cap evolution in the 21st century

Our dynamical model simulations reveal that Jostedalsbreen stays in near balance until year 2040. Thereafter, the ice cap is projected to lose half (two-thirds) of its ice volume under medium (high) emissions until the year 2100 (Figs. 2d and 9b). These results are obtained with the ECEARTH/CCLM climate forcing, which gives an ice-volume loss that is mid-range across the four climate model combinations (Fig. 11; cf. Table 2). Model simulations forced by the full ensemble of climate models suggest that, until the year 2100, Jostedalsbreen will lose 10–75 % of its present-day volume under medium emissions (Fig. 11f). Meanwhile, for high emissions, future volume evolution ranges from 75% loss (CNRM/CCLM) to 17% gain (MPI/CCLM) by 2100. The latter unexpected future increase of volume with MPI/CCLM under high emissions is a result of positive SMB. While we include these results for completeness, we deem this climate projection unrealistic (Sect. 6.2.1). If we exclude the

MPI/CCLM results, the projected future mass loss is 12–74% for medium emissions, and 53–74% under high emissions (Fig. 11f).

In our experiments of future evolution until 2100, Jostedalsbreen splits up into three separate ice caps both under medium and high emissions (ECEARTH/CCLM, RCP4.5 and 8.5; Fig. 6). With medium emissions, North separates from the rest of the ice cap in the 2070s, while South becomes a separate ice cap in the 2090s (see Supplementary video 1). Meanwhile, under high emissions, these breakups occur around 10–20 years earlier; in the 2050s for North, and in the 2080s for South (see Supplementary video 2). However, it is important to note that these separations and other small-scale details include a high degree of uncertainty. They can vary by several decades due to differences in the future climate projections (Sect. 7.1), underlying model limitations (Sect. 7.3) and local mismatches between the modelled and real-world geometry. For example, the model mesh is too coarse (Fig. A1) to resolve a few small ice-free patches (100–200 m across) which currently partly separate the southern and central parts (cf. Fig. 1). Modelled contemporary ice thickness is 20–50 m in this area, while the measured ice thickness is less than 25 m (Gillespie et al., 2024). This suggests that a separation may occur several decades before the 2080s. The North part of Jostedalsbreen, with its plateau situated c. 100–300 m lower than the Central ice cap (Fig. 1), is simulated to decline most strongly. The modelled future thinning here is 20–100 %, and total volume loss 74–93%, depending on emissions (Fig. 6). This is already the area of the ice cap where current mass loss is greatest (Andreassen et al., 2023; Sjursen et al., 2025).

Regardless of emission pathway, most of the volume loss is driven by strong thinning of 20–200 m at the ice-cap margins (Fig. 6), with less pronounced thinning of 20–100 m on the plateau. This results in frontal retreat and an overall steepening of the ice cap's outlet glaciers (Fig. 7ab). These patterns can be explained by a SMB<sub>transGeom</sub> that, over the course of the century, becomes increasingly negative in ablation zones, while net annual SMB on the plateau remains positive for many of the years in the coming decades (Fig. 7cdf). Annual SMB on the plateau is in fact rather similar between the medium and high emissions, which is likely due to similar precipitation amounts (see Sect. 7.1). Meanwhile, SMB along the margins of key outlet glaciers such as Nigardsbreen and Tunsbergdalsbreen is 3–8 m w.e. more negative than at present (Fig. 7cd). This corresponds to typical annual SMB values of -12 to -18 m w.e. at the frontal regions, which presently are located at 300-700 m a.s.l.

Depending on the emission scenario, terminus retreat for Nigardsbreen and Tunsbergdalsbreen is simulated to be c. 2.5–5 km and c. 5–10 km, respectively (ECEARTH/CCLM; Fig. 7ab). Because mass loss at the high-lying plateau here is rather subdued, these glaciers still retain around half of their volume by the year 2100, even under high emissions (Fig. 8). Similarly, Austerdalsbreen loses the entire glacier tongue below its ice fall under RCP4.5, but two-thirds of the ice volume remains by 2100. To reiterate this point: Tunsbergsdalsbreen retreats 10 km under high emissions (Fig. 7b), yet by the end of the century still hosts large upper areas with ice more than 500 m thick (Fig. 6d). As we will see in Sect. 6.2.3, this is the most resilient part of the ice cap.

## 6.2.3 High committed mass loss 2100–2300






Our simulations show that the ice cap is in strong disequilibrium with the current climate. If the climate of the last 20 years (2000–2020) would persist into the future, almost half (47%) of the ice-cap volume would be lost (CommitNow; Figs. 9b)

**Figure 6.** Future model thickness and thickness change under medium (a–c) and high (d–f) emissions, using ECEARTH/CCLM climate forcing. (a,d) Modelled ice thickness in year 2100; (b,e) Total ice thickness change (future-present); (c,f) Relative thickness change. In (e–f), yellows means a slight surface lowering, while dark red implies strong thinning. In (c,f), -100 % thickness change means complete deglaciation

**Figure 7.** Elevation of bedrock and modelled ice-surface evolution (a, b and e) and SMB evolution (c, d and f) 2021–2100 under RCP4.5 (ECEARTH/CCLM) for three transects on Jostedalsbreen. Panels (a, b and e) illustrate a consistent, progressive glacier surface lowering over time, despite some decadal variability in surface mass balance (c, d and f). The three transects are shown in the inset map, which is slightly rotated for visibility.

**Figure 8.** Future mass loss for South, Central and North parts (delineated by dashed lines) and selected glacier basins (thick coloured outlines) of Jostedalsbreen ice cap. The discs show relative mass loss (% of 2020 volume) and remaining volume (km³) in year 2100 (RCP4.5, RCP8.5) and 2300 (Commit4.5, Commit8.5). Modelled present-day ice volume is shown next to each disc. The area of discs for Jostedalsbreen, South, Central and North are scaled by their 2020 volumes. The area of individual glacier basins are scaled similarly (not to scale with the regions). The background map shows simulated thickness in year 2100 under scenario RCP4.5, and glacier outlines in 2019 (black lines Andreassen et al., 2022). Climate forcing is based on ECEARTH/CCLM (complete list in Table 2).

and 10a). Under the medium-emission scenario RCP4.5, Jostedalsbreen loses around half of its present-day volume by 2100. Similarly, about one-third of the ice volume remains by 2100 under the high-emission scenario RCP8.5 (Fig. 9b). However, this is only half of the story, since Jostedalsbreen will not have reached a steady-state in 2100. Further mass loss is 'in the pipeline', as the ice cap undergoes additional dynamic adjustments and continues to thin due to the effective SMB-elevation feedback. In the high-emission case, our simulations show that the ice cap is committed to vanish completely by around 2200, even without any post-2100 climate warming (Commit8.5; Figs. 8 and 9b). Similarly, with moderate 21st-century emissions (RCP4.5), the ice cap is committed to lose 90% of its present-day volume by 2300 (Commit4.5). Upper Tunsbergsdalsbreen and the nearby high-elevation areas to the north (upper Brenndalsbreen and southwest corner of upper Nigardsbreen) is the part of the ice cap that is most likely to survive the longest. Another resilient high-lying ice complex appears to be upper Fåbergstølsbreen and nearby areas along the local ice divides (Fig. 10bc).

## 6.2.4 Failure to recover





Our Reverse experiments test whether the ice cap can recover if the climate is reversed back to present immediately at the year 2100. That is, instantly resetting the climate forcing to an SMB associated with the prevailing climate of 2000–2020. We find that such hypothetical time travel leads to barely any recovery (Fig. 9b). The inability to recover holds regardless of whether emissions are moderate or high before the reversal. The main reason for the failure to recover is that the ice cap is currently strongly out of balance with the prevailing climate. In other words, resetting the climate of year 2100 to the climate of 2000–2020 means aiming for an ice-cap that is considerably smaller than that of the present-day. The CommitNow experiment illustrates this, where Jostedalsbreen's volume is halved by the year 2300 (Fig. 9b). Indeed, by 2300, the Reverse4.5 and Reverse8.5 simulations would reach steady-state volumes similar to that of the CommitNow experiment, if the simulations are allowed to continue for another 100 years (c. 30–35 km³; Fig. 9b). Another aspect preventing recovery may be that by 2100, the ice cap is on the course to an even lower ice volume (cf. Commit4.5 and Commit8.5 experiments; Fig. 9b). Therefore, any ice-cap growth due to a more favourable SMB, will in fact compete with ongoing dynamic adjustments and associated SMB-elevation feedback working towards a smaller ice cap.

The found failure to recover, as well as the difficulty to regrow the ice cap (Sect. 7.2), is also an aspect of the hysteresis effect (Oerlemans, 1981). This hysteresis is particularly strong for ice caps and ice sheets (Garbe et al., 2020), since a large part of their area is concentrated within a small elevation range. These ice masses can be maintained as long as sufficiently large areas with positive SMB persist. If the ELA rises above such flat high-elevation areas, mass loss quickly accelerates and becomes difficult to reverse (Giesen and Oerlemans, 2010; Åkesson et al., 2017).

**Figure 9.** Historical and future evolution of (a) reference SMB (SMB<sub>static</sub>), (b) ice volume, and (c) area, using future ECEARTH/CCLM climate forcing. Historical observations (gray circles) and uncertainties (whiskers) are shown. See Table 2 for a full experiment overview.

**Figure 10.** Maps of modelled ice thickness in 2300, after accounting for committed mass loss due to (a) SMB 2000–2020, (b) medium 21st-century emissions; (c) high 21st-century emissions. These 'Commit' simulations have temporally fixed SMB, and run over (a) 2021–2300; (b) 2101–2300; and (c) 2101-2300, see Table 2. The SMB in (b) and (c) is derived from ECEARTH/CCLM climate forcing. Present-day glacier outlines (2019) are shown with black lines (Andreassen et al., 2022).

## 7 Discussion




## 7.1 Future surface mass balance

Sensitivity to climate models. The majority of our simulations suggest that Jostedalsbreen will experience a substantial mass loss over the course of the 21st century (Fig. 11). At the same time, modelled SMB based on the ensemble of climate models leads to a large spread of possible future ice-cap evolution. This suggests both substantial variations between projected temperature and precipitation changes between climate models (Fig. A5) and a strong sensitivity to choice of climate model and scenario. Medium and high emission scenarios show a mean annual temperature increase over Jostedalsbreen (2019 area) of 1.8 °C and 3.4°C, respectively, by the end of the century. The mean increase in annual precipitation sums for the respective scenarios is 27% and 65%, with an increase of 30% and 67% in winter (Fig. A5c). However, for a given emission scenario there are considerable differences between the models. For example, projected change in summer (May–Sep) temperature varies from -0.3°C to +2.7°C under RCP4.5, and +0.2°C to 3.0°C under RCP8.5 (Fig. A5b). The spread in winter (Oct–Apr) temperature is smaller, around 1.2°C and 1.9°C for RCP4.5 and 8.5, respectively (Fig. A5a). Differences in precipitation between climate models and scenarios are even more striking, with models projecting winter precipitation changes between -14% and +72% for

RCP4.5 and between +39% and +80% for RCP8.5 (Fig. A5d). Moreover, the timing of precipitation changes also varies across the climate models, both in terms of the time of year and the evolution throughout the century.

Seasonality. Our simulations allow us to assess future changes in seasonality and magnitudes of ablation and accumulation for Jostedalsbreen. With the ECEARTH/CCLM projections, the largest changes in the SMB for the transiently evolving Jostedalsbreen can be expected in autumn and summer under medium and high emissions, respectively (Fig. 12). Under an RCP4.5 scenario, the ablation season is prolonged by approximately a month, from mid-September to mid-October, with an equivalent shortening of the winter accumulation period (Fig. 12a). Meanwhile, ablation magnitudes at the peak of summer change relatively little. This contrasts with high emissions, where strong summer melt intensifies and is more prolonged between June to September (Fig. 12b). These differences reflect both differing warming and precipitation trends between climate projections (Fig. A5), and variations in which seasons and months such changes are projected to occur. While both ECEARTH/CCLM scenarios render relatively strong winter warming (Fig. A5a), temperature increase in summer is smaller (Fig. A5b). Moreover, there are variations between the projections within seasons. For example, warming is stronger in October for RCP4.5 than RCP8.5, which could explain why autumn SMB changes are less pronounced for the latter. However, it should be noted that the future monthly SMB in Fig. 12 not only reflects climatic changes but also accounts for the feedback of glacier retreat and surface-elevation lowering (SMB<sub>transGeom</sub>).

**Precipitation.** Most climate models show that precipitation gains increase with elevation. This explains why the models indicate larger precipitation increases over Jostedalsbreen than for western Norway in general (Hanssen-Bauer et al., 2017). This may be a natural consequence of increasing and intensifying precipitation; precipitation amounts on Jostedalsbreen are likely substantially affected by orographic uplift (e.g. Ketzler et al., 2021). However, the large precipitation increase over Jostedalsbreen may also be a result of how climate models are downscaled and bias-corrected to the seNorge version 1.1. dataset (Wong et al., 2016), which relies on precipitation lapse rates to extrapolate precipitation to higher elevations (Mohr, 2008). The large projected increases in precipitation, and differences between climate models and scenarios, highlight the considerable uncertainty in future temperature and precipitation in mountainous regions, and the sensitivity of glacier projections to climate model input. For glaciers with a substantial mass turnover and likely large future accumulation area, such as Jostedalsbreen, our results highlight that future glacier evolution strongly depends on future precipitation changes, and that projections of future glacier change remain uncertain due to the (in)ability of climate models to capture these changes.

SMB feedbacks. In future SMB simulations there are two competing feedbacks between SMB and glacier geometry changes at play (Elsberg et al., 2001; Huss et al., 2012). The first is the retreat feedback, which contributes to a more positive SMB<sub>transArea</sub>, relative to the reference SMB<sub>static</sub>, as the glacier retreats to higher elevations where ablation is smaller. The elevation feedback acts in the opposite direction, contributing to a more negative SMB<sub>transGeom</sub> as the glacier surface is lowered to elevations where ablation is higher (Huss et al., 2012). To quantify the relative and net impacts, we computed the transient SMB in Megatonnes (Mt) across the entire ensemble of climate model combinations (Fig. 11d). This analysis shows that, for most climate models, the total mass balance SMB<sub>static</sub> becomes very negative towards the end of the century, since it ignores the two feedbacks. Meanwhile, the positive retreat feedback, which includes area changes, makes the SMB<sub>transArea</sub> c. 0.5–2.5 Mt less negative than SMB<sub>static</sub>. Note that the glacier is still losing mass overall for all scenarios with this feedback included.

Including also the elevation feedback, where glacier geometry and SMB is fully coupled (SMB $_{transGeom}$ ), shifts the total mass balance 0.1–0.6 Mt more negative compared to SMB with the retreat feedback only (SMB $_{transArea}$ ). Simulation of long-term glacier evolution thus requires careful consideration of the mass-balance elevation feedback. If not, a strong positive bias in modelled SMB can be expected (Fig. 11).


**Figure 11.** Future evolution of SMB and ice volume 2021–2100 across the ensemble of climate model forcings, cf. Table 2. (a) SMB with fixed present-day ice geometry (SMB<sub>static</sub>); (b) SMB over the modelled area evolving over time, without accounting for the SMB-elevation feedback (SMB<sub>transArea</sub>; see Sect. 4.3); (c) SMB over a transiently updated area and elevation, (SMB<sub>transGeom</sub>); (d) total SMB of the representations in (a–c) in Megatonnes ( $10^6 \text{ m}^3 \text{ w.e.}$ ), using a 5-year moving mean; (e) Cumulative SMB<sub>transGeom</sub>; (f) Modelled ice volume with SMB<sub>transGeom</sub>. Line styles shown in the legend in (f) are consistent in (a–c and e–f), representing climate model combinations ECEARTH/CCLM (ECC), ECEARTH/HIRHAM (ECH), CNRM/CCLM (CNC), and MPI/CCLM (MPC).

**Figure 12.** Mean monthly surface mass balance (SMB) for historical (SMB<sub>static</sub>, grayscale; fixed DTM 2020 geometry and modelled 2020 area) and future (SMB<sub>transGeom</sub>; colours) decades for ECEARTH/CCLM (a) RCP4.5 and (b) RCP8.5. The 2090s mean refers to the 11-year period 2090–2100.

## 7.2 Jostedalsbreen's fate compared to other glaciers in Norway




**Potential for regrowth.** A pertinent question is whether Jostedalsbreen can regrow from no ice to its current size under contemporary or future climate conditions. The answer is no, neither in the current climate, nor in a warmer future one, at least not using the chosen main climate model combination ECEARTH/CCLM. If Jostedalsbreen disappeared tomorrow, the present-day climate (2000–2020) would render a much smaller ice cap than the present-day Jostedalsbreen, with a 59% smaller ice volume (29 km³), 30% smaller area (317 km² and 60% thinner ice [mean ice thickness of only 91 m]; Fig. A3). This implies 60% thinner ice on average than for the present-day (Gillespie et al., 2024). The modelled volume of this regenerated ice cap is similar to present-day Søndre Folgefonna (30 km³; Ekblom Johansson et al., 2022), Norway's third largest ice cap by area, and around three times larger than the present-day Hardangerjøkulen (ice volume of 9.3 km³, area of 71 km², max ice thickness c. 350 m), one of the most well-studied ice caps in Norway (Andreassen et al., 2015). Meanwhile, we barely produce any glacier ice when trying to grow Jostedalsbreen from no ice using an SMB (2081–2100) associated with medium 21st-century emissions, and no ice at all with a high-emission climate (experiments Regrow4.5 and Regrow8.5). For Hardangerjøkulen, Åkesson et al. (2017) similarly found that a small negative SMB anomaly relative to the present-day would render a regrowth from no ice impossible. These findings collectively imply that if two of the largest Norwegian ice caps would melt away, a recovery appears very difficult, and would require a colder and/or wetter climate than the present one.

**Large-scale studies.** Regional studies of Scandinavian glaciers have shown that Jostedalsbreen will lose 35% (RCP4.5) to 50% (RCP8.5) of its area and volume by 2100 (Huss and Hock, 2018). This is slightly lower mass loss than estimated by the

current study (49% for RCP4.5, 63% for RCP8.5; Fig. 8). Meanwhile, Compagno et al. (2021) suggested that Nigardsbreen would lose 64% (RCP4.5) to 85% (RCP8.5) of its present-day volume, which is roughly 50% more mass loss than our midrange results for Nigardsbreen from ECEARTH/CCLM (Figs. 8). More broadly, estimates for 21st-century mass loss for Scandinavian glaciers and ice caps as a whole range from 40–80% (RCP4.5) and 70–100% (RCP8.5; e.g. Marzeion et al., 2012; Compagno et al., 2021; Rounce et al., 2023). These large-scale studies may perform well on regional scales, but their estimates for individual ice masses are not directly comparable with those of a detailed study like ours. As emphasised in Sect. 1, regional studies treat connected glacier units separately, use crude ice-flow dynamics, and have not benefitted from such detailed ice-thickness data and surface mass balance simulations as we have in the current study. The methods, scales and implementations thus differ greatly compared to the detailed, higher-order, high-resolution, spatially continuous modelling approach presented here.

**Detailed studies.** For high-emission scenarios, some studies of other ice caps in Norway suggest almost complete disappearance by the year 2100 or soon thereafter, including Hardangerjøkulen (Table 3; Giesen and Oerlemans, 2010; Åkesson et al., 2017), and Spørteggbreen ice cap, c. 15 km east of Jostedalsbreen (Laumann and Nesje, 2014). Our findings suggest that Jostedalsbreen will be more long-lasting, and retain one-third to half of its present-day ice volume at the end of the 21st-century, depending on emissions (Fig. 9b). Not only does Jostedalsbreen contain more ice – its particular resilience is likely also related to the vast plateau being at high elevations, and relatively large precipitation amounts, which will increase even more in the future. Even so, an extrapolation of the high-emission modelled rate of mass loss by 2100 suggests a complete disappearance in the late 22nd-century, if emissions are not curtailed. While our Commit-experiments do not explicitly test this, they illustrate that by 2100, Jostedalsbreen may be committed to a complete disappearance 200 years later.

Oerlemans (1997) simulated future evolution of Nigardsbreen (cf. Fig. 1) using a 1d-flowline model. For a warming of 1 °C and +10% precipitation, Nigardsbreen was projected to lose 24% of its present-day (AD 1950) volume, while for a 2 °C-warming and +20% increase in precipitation, Nigardsbreen's modelled future volume loss was 58%. This is comparable to our modelled 40–56% volume loss for Nigardsbreen for RCP4.5 and RCP8.5, respectively (Fig. 8). However, the thickness data, models and future climate projections used by the current study and Oerlemans (1997) differ significantly.

The impact of future warming on Briksdalsbreen has previously been modelled using a flowline model together with a degree-day model for surface mass balance (Laumann and Nesje, 2009). While they did not provide volume projections, they estimate 21st-century frontal retreat of 2.5 to 5 km under scenarios similar to RCP4.5 and RCP8.5 used here. A similar retreat is simulated in the current study, along with a volume loss of 33% (RCP4.5) and 46% (RCP8.5) for the glaciers terminating in Oldedalen, using the ECEARTH/CCLM climate (Fig. 8).

Results from Søndre Folgefonna indicate that for RCP4.5 and RCP8.5 about 43% and 92%, respectively, of the ice cap will have melted away by the year 2120 (Ekblom Johansson et al., 2022). The northern parts of the ice cap are more vulnerable than the southern parts. Søndre Folgefonna's current ice volume is 28 km³ (Ekblom Johansson et al., 2022), around 40% smaller than the ice volume of Jostedalsbreen. Considering its smaller size, Søndre Folgefonna is surprisingly resilient to future warming. The knowledge of the bed topography underneath Søndre Folgefonna (Ekblom Johansson et al., 2022) is perhaps the only one in Norway that matches the level of detail in the newly-mapped topography under Jostedalsbreen (Gillespie et al., 2024), and

**Table 3.** Future volume loss of glaciers and ice caps in Norway, based on this and previous studies. For simplicity, we associate low, medium and high emissions with the respective RCP scenarios, although all studies do not strictly adhere to these scenarios. Shown are present-day volumes (varying year according to publication), and change in ice volume ( $\Delta V$ , %) in projections to year 2100.

| Glacier            | Volume (km <sup>3</sup> ) | low emissions<br>RCP2.5<br>(\(\Delta\V\), %) | medium emissions RCP4.5 $(\Delta V, \%)$ | high emissions<br>RCP8.5<br>(ΔV, %) | Reference                      |
|--------------------|---------------------------|----------------------------------------------|------------------------------------------|-------------------------------------|--------------------------------|
| Jostedalsbreen     | 70.6                      |                                              | -49                                      | -63                                 | This study                     |
| Oldedalen glaciers | 8.4                       |                                              | -33                                      | -46                                 | This study                     |
| Tunsbergdalsbreen  | 10.8                      |                                              | -41                                      | -51                                 | This study                     |
| Austerdalsbreen    | 3.7                       |                                              | -34                                      | -49                                 | This study                     |
| Nigardsbreen       | 7.4                       |                                              | -40                                      | -56                                 | This study                     |
| Nigardsbreen       | 7.4                       | -24**                                        | -42**                                    |                                     | Oerlemans (1997)               |
| Briksdalsbreen     | 2.0                       |                                              | 2.5 km retreat                           | 5 km retreat                        | Laumann and Nesje (2014)       |
| Hardangerjøkulen   | 9.3                       | -64                                          | -84                                      | -97                                 | Giesen and Oerlemans (2010)    |
| Hardangerjøkulen   | 9.3                       |                                              | -90–100                                  | -100                                | Åkesson et al. (2017)          |
| Søndre Folgefonna  | 28                        |                                              | -43*                                     | -92*                                | Ekblom Johansson et al. (2022) |
| Spørteggbreen      | 2.5                       |                                              | -30                                      | -90–100                             | Laumann and Nesje (2009)       |

<sup>\*:</sup> State in year 2120. \*\*: relative to volume in AD 1950




maximum ice thicknesses are similar, with areas of ice more than 500 m thick existing for both ice caps. Some outlet glaciers from Folgefonna are known to have reached their Little Ice Age maximum extent as late as in the 1940s (Tvede, 1973; Tvede and Liestøl, 1976), and has not retreated much since, which may explain the relative resilience to warming. Folgefonna is also a more maritime ice cap than Jostedalsbreen, with its surface mass balance heavily controlled by high winter snowfall (Tvede, 1989). This may buffer Folgefonna from future warming, and perhaps also the western outlets of Jostedalsbreen. Such snowfall buffering however has its limits. As climate warms, more precipitation falls as rain rather than snow, which, for example, has been found to diminish the buffering of Patagonian glaciers (Troch et al., 2024) located in a similar maritime climate (Tvede, 1973).

#### 7.3 Limitations and future improvements

The agreement between our model system for ice dynamics and SMB and reality depends on a multitude of model choices and assumptions, as well as uncertainty in observational data. These include uncertainty in historical and contemporary ice-surface data and bedrock topography (Sect. 3.1), uncertainty in velocity data (Sect. 3.2), ice-flow physics, basal friction, ice rheology, and mesh construction (Sect. 4.1), surface mass balance model and its parameters (Sect. 4.3), as well as uncertainties in climate projections (Sect. 7.1). Most of these factors also vary spatially across the 81 glacier units.

Input data. The gridded observation-based ice-thickness dataset is based on measured ice-thickness point data combined with an ice-thickness model based on inversion (Sect. 3.1; see Gillespie et al. (2024) for details). Despite the unprecedented level of detail in this dataset, the uncertainties in areas without thickness observations are estimated to be up to 50–100 m, but in practice likely smaller, particularly close to radar survey lines (Gillespie et al., 2024). An overestimated (underestimated) observation-based thickness will affect modelled ice-flow and render a too slow (too fast) future mass loss and retreat. Meanwhile, present-day thickness and ice-cap margins are overall well reproduced by the historical simulations, yet there are some mismatches at the end-of-historical state (Fig. 3). These differences for individual glaciers will propagate into the future, affecting the timing of retreat. For example, the front positions of the steep Briksdalsbreen and Brenndalsbreen outlets are c. 700 and 1200 m too advanced, respectively, for the present-day (cf. Figs 1 and 3a). This disagreement likely propagates several decades into the future simulations. This means that future modelled frontal positions for these glaciers are likely located several hundred metres further downvalley compared to reality. Meanwhile, the fronts of these and other outlet glaciers with steep ice falls (e.g. Austerdalsbreen) are thin and may disconnect from the upper glacier. This is modelled for Brenndalsbreen (Fig. 7b) and illustrated for similar ice masses (Davies et al., 2024). In this case, the disconnected tongue will lose its ice supply from the upper glacier, and quickly melt away.








**Initial state and spinup.** The future projections presented here rely on the model geometry and dynamics of the historical simulations 1960-2020. In turn, the historical experiments are influenced by the initial ice-cap state of the 1960s, obtained through model spinup (Sect. 5.1). The spinup uses the 1966 ice-surface topography as a target, and assumes that the ice cap was in steady-state in the 1960s. We also tested to obtain the initial historical geometry without a steady-state assumption, starting from the 1966 surface DEM and running a relaxation for only a few years. While ice-surface data artifacts dissipated, this approach resulted in unrealistic model drift of the ice-surface topography during the historical simulation. We therefore switched to the practical steady-state assumption. An inverse approach to find the ice thickness (Frank and van Pelt, 2024) could also have been considered, yet here we take advantage of the detailed newly collected thickness dataset from Gillespie et al. (2024). In reality, the ice cap will have experienced ongoing adjustments to climate variations occurring between the Little Ice Age maximum in the 1800s up until the 1950s. Indeed, several records of frontal variations of Jostedalsbreen's outlet glaciers show retreat in the 1930s through 1960s (https://glacier.nve.no/viewer/CI/en/; Andreassen and Elvehøy, 2021; NVE, 2025). Such adjustments are not accounted for in the historical simulations. This may partly explain why the model underestimates the reconstructed ice volume in 1966 (Fig. 2a), and partly why a relatively modest historical retreat and volume loss is simulated – although we emphasise that there are also underlying uncertainties in the observation-based ice thickness dataset (Gillespie et al., 2024). Regardless, not accounting for the real-world ongoing decline could have resulted in chosen model parameters being biased towards too positive SMB.

**SMB and climate.** Historical and future ice cap evolution is strongly controlled by the SMB forcing (Figs. 2 and 9), which is subject to its own limitations, in addition to the uncertainty in climate forcing. There are several features that our SMB model does not explicitly account for, including spatially variable processes such as redistribution of snow by wind and avalanching, and differences in incoming solar radiation with different aspects and topographic shading. However, some of these characteristics could be accounted for by the spatially variable model parameters (see Sjursen et al. (2025) for a detailed

discussion). We have also assumed that parameter values in the SMB model are constant over time, including in a changing future climate. Some studies indicate that this may cause an oversensitivity to temperature change (Gabbi et al., 2014; Ismail et al., 2023), although it is unclear if more physically-based approaches improve the performance when detailed meteorological data is lacking (Réveillet et al., 2018). Nevertheless, the above discussion in Sect. 7.1 illustrates that modelled future SMB, and thus ice-cap evolution, strongly depends on the choice of climate model, where especially precipitation is uncertain. Thus, the uncertainty in future climate forcing, illustrated by the large spread in future projected temperature and precipitation (Fig. A5), likely outweighs the uncertainty associated with SMB model parameter values. In our Commit-experiments, the mean SMB 2081–2100 is used as a constant climate forcing. Other time periods could be considered to represent the end-of-the-century climate, but the mean SMB over two decades was chosen in order to be consistent with the two-decade mean for present-day SMB in the CommitNow experiment.







**Ice dynamics.** Modelled ice dynamics is heavily influenced by the representation of basal friction (e.g. Brondex et al., 2019; Åkesson et al., 2021). Here we have taken advantage of a global velocity dataset for the fastest outlet glaciers. The velocity data appears to be of good quality for the large outlet glaciers (Sect. 3.2), and more uncertain for small and narrow glaciers. Comparison against the global velocity dataset suggests that several model glaciers are less dynamic compared to observations (Fig. 3d). Given accurate velocity data, this could partly explain why historical outlet-glacier retreat is underestimated in a few cases (e.g. steep glaciers such as Briksdalsbreen and Brenndalsbreen; Fig. 1). It follows that more dynamic outlet glaciers would render faster frontal retreat than what we have projected here.

The global velocity dataset used does not come without issues, as highlighted in Sect. 3.2. We illustrated a hybrid approach to mediate this, with inversion for the largest and most dynamic glaciers, where we judge the velocity data quality to be highest. While the quality of the velocity data varies spatially, it also provides limited temporal information. This means that friction coefficients are uncertain over the modelled historical timespan, let alone in future simulations. Like virtually all previous model studies, we keep the friction parameters fixed in time, ignoring any feedbacks between increased meltwater supply and friction, which in a future warmer climate may accelerate mass loss further (e.g. de Fleurian et al., 2022).

Several alternative approaches to basal friction were tested. We tried a transient inversion of the friction field, where ice thickness was allowed to evolve using a constant SMB, and the friction parameter adjusted at regular intervals (10–50 years) depending on local thickening or thinning, following Pollard and DeConto (2012); Santos et al. (2023). Despite theoretical promise, this approach resulted in large unrealistic drift of the ice-cap geometry during spinup, and was therefore abandoned. Moreover, we also tested a spatially uniform friction parameter, an elevation-dependent friction parameter everywhere (Eq. 3), as well as imposing the friction inversion everywhere (Eq. 2). All of these approaches rendered poorer results (higher RMSEs) than our approach outlined in Sect. 4.2. For example, using a friction inversion for the entire ice cap (Eq. 2; Sect. 4.2), increases the mismatch with present-day observed ice volume from 0.4% using the current hybrid approach to 3.5% using the inversion everywhere.

The spatially variable friction field obtained aims to represent the bulk effect of all interactions between ice, water, rock and sediments at the ice-bed interface. Still, friction parameter adjustments are essentially also compensating for potentially missing physics and uncertainties in the bed topography, ice-flow physics and surface mass balance forcing. In this light, we

expect and accept some mismatches between the modelled and observation-based thickness, as well as between modelled glacier fronts and their observed historial and present-day positions.

## 7.4 Global implications

Glacierised regions worldwide are in decline and are on track for severe mass loss over the 21st-century. If emissions are not strongly reduced, many glaciers are bound to recede or disappear, including in the European Alps (e.g. Jouvet et al., 2011). High-Mountain Asia (Van Tricht and Huybrechts, 2023); the sub-Antarctic (Verfaillie et al., 2021) and Scandinavia (Giesen and Oerlemans, 2010). Numerous smaller mountain glaciers may disappear even under the 1.5°C goal of the Paris Agreement (e.g. Rounce et al., 2023). Large ice caps like Jostedalsbreen contain more ice and will therefore take longer to disappear. Ice masses with flat-topped geometry and flow governed by topography are usually called ice caps in Scandinavia, Iceland, Greenland, and Canada, while referred to as 'ice fields' in Alaska and Patagonia. Studies from Alaska (Ziemen et al., 2016), Arctic Canada (Schaffer et al., 2023), Iceland (Schmidt et al., 2020), Greenland (Zekollari et al., 2017) and Patagonia (Troch et al., 2024) illustrate that large ice caps and ice fields may last several hundred years into the future even under strong warming. This does not mean that ice caps are necessarily more resilient in a warming climate – rather the opposite. Because of their flat interior plateaus, ice loss will accelerate greatly as the ELA rises to and above the elevation of vast plateau areas (Harrison et al., 2001; Åkesson et al., 2017). This switch to a negative SMB over large areas drastically reduces the accumulation area – a recipe of strong mass wastage and eventual disappearance. In the case of Jostedalsbreen, our simulations suggest that this situation occurs towards the end of the 21st-century (Fig. A4c and f). Under high emissions, the associated accumulation-area ratio for 2081–2100 is approximately 0.1, meaning that 90% of the ice-cap area experiences net annual melt (Fig. A4e and f). The strong long-term impact is illustrated with the Commit8.5 experiment, which leads to a 99% volume reduction of Jostedalsbreen by the year 2300 (Figs. 8 and 11c). Under the more moderate emission pathway RCP4.5, areas with net positive SMB by the end of the 21st century are somewhat larger (Fig. A4), yet Jostedalsbreen is at that stage committed to a mass loss of 90% of its present-day volume (Figs. 8 and 11c).

## 8 Conclusions and outlook

Jostedalsbreen is the largest ice mass in mainland Europe. The state and fate of this and similar ice masses worldwide is of great interest to local residents and stakeholders because of widespread natural and socio-economic implications.

To simulate the future evolution of Jostedalsbreen, we use a 3-D ice-flow model coupled with a surface mass balance model until 2100 and beyond to 2300, allowing us to investigate rapid geometric changes in complex terrain. Historical simulations with the ice-flow model from 1960 to 2020 serve to reproduce present-day conditions in terms of ice geometry and dynamics. This current-state ice cap then provides a vantage point for embarking on annually coupled ice-dynamical and mass balance simulations, fully accounting for the mass-balance elevation feedback.

 We find that Jostedalsbreen is subject to substantial mass loss until 2100, losing about half of its 2020 volume under the medium greenhouse-gas emission scenario RCP4.5, and about two-thirds under the high-emission scenario RCP8.5. There exists a large spread in simulated mass loss ranging from 12–74% (RCP4.5) and 53–74% (RCP8.5), depending on the employed climate model, while excluding results from one model which generates unrealistically high future winter accumulation. Focusing on results from the mid-range results obtained using the model ECEARTH/CCLM, Jostedalsbreen is projected to lose 49% (RCP4.5) and 63% (RCP8.5) of its present-day ice volume by 2100.

- Most simulations suggest that Jostedalsbreen will split into three separate ice caps over the course of the 21st century.
  The ice cap is however currently *not* (yet) committed to this breakup. This implies that if greenhouse-gas emissions are reduced over the next decades, the ice cap may evade future breakup and complete disappearance.
- Simulations until the year 2300 show that with the current climate, Jostedalsbreen is committed to losing almost half of its current volume by the year 2200. Similarly, prolonging the surface mass balance representative for medium- (RCP4.5) and high emissions (RCP8.5) at the end of the 21st century (2081–2100) suggest that the ice cap is bound to shrink by 90% until the year 2300, or disappear by 2200, respectively.
- Tunsbergdalsbreen, the largest glacier basin, appears to be most resilient to future climate change. Its upper reaches are likely to persist well beyond 2100, with 500 m thick ice still present by 2100.
  - Experiments beyond 2100 with the surface mass balance instantly reset to that of the present-day indicate that large parts of mass losses until 2100 are irreversible. This is due to significant ice-surface lowering and the associated strong mass-balance elevation feedback, which results in SMB being about 2 m w.e. more negative than that observed in 2001–2020.
- We have demonstrated the importance of accounting for the mass-balance elevation feedback when assessing ice-cap evolution over long timescales. This feedback partly explains why Jostedalsbreen would likely not regrow to a large extent if it disappeared. The current climate of the 2000s is not favourable enough to regenerate the contemporary ice cap.

Jostedalsbreen is likely to survive longer under future warming than many smaller glaciers and ice caps in Scandinavia and worldwide. Future climate conditions determine the SMB over the ice cap, and appear to be the largest source of uncertainty in our projections. Our findings indicate that climate models particularly disagree on future precipitation changes, which affects glacier projections of maritime ice caps like Jostedalsbreen. This calls for further improvement of regional climate models and the representation of precipitation in mountainous areas.

Our detailed, spatially continuous model approach for an ice mass with 81 glacier units, with challenging topography and ice-flow regime, should serve as inspiration for studies of glaciers and ice caps worldwide. Numerical simulations of complex ice masses like Jostedalsbreen will also serve as critical training data for more robust and widely applicable emulators based on machine learning (e.g. Jouvet et al., 2021; Jouvet and Cordonnier, 2023). The diverse characteristics make ice caps ideal yardsticks for such emulators in future studies. Finally, our findings on strong committed mass loss illustrate the need to consider time scales beyond the 21st century, and call for intensified research on irreversible change and committed mass loss.

Code and data availability. ISSM is open-source and can be downloaded from https://issm.jpl.nasa.gov. Model code to run the simulations, as well as simulated future surface mass balance, are available at https://github.com/henninma/jost (https://doi.org/10.5281/zenodo.17304935).
 Climate model projections for 2021–2100 can be downloaded from the Norwegian Center for Climate Serviceswebsite (https://klimaservicesenter.no/; Jacob et al., 2014; Wong et al., 2016). Data for present-day ice thickness and bed topography are available for download at https://doi.org/10.58059/yhwr-rx55 (Gillespie et al., 2024). Land- and ice-surface DTM 2020 is available from https://hoydedata.no/. The 1966 ice-surface is available from
 [We are making this available. To be added at the proof-reading stage.] The 2019 glacier outlines are available at https://nve.brage.unit.no/nve-xmlui/handle/11250/2836926. The 1966 outlines are available at https://doi.org/10.58059/7yte-3c61. Data of glacier front position changes are available at https://glacier.nve.no/Glacier/viewer/CI/no/. Global ice-velocity data are available at https://doi.org/10.6096/1007 (Millan et al., 2022), and the 1996 ice-velocity data is available at [DOI link to be added].

*Video supplement.* The following animations are supplied along with the article, and can be downloaded from https://github.com/henninma/jost (https://doi.org/10.5281/zenodo.17304935).

- Supplementary video 1: Modelled ice-thickness evolution 2021–2100 (ECEARTH/CCLM, RCP4.5).
- Supplementary video 2: Modelled ice-thickness evolution 2021–2100 (ECEARTH/CCLM, RCP8.5).
- Supplementary video 3: Modelled ice-thickness evolution 2101–2300 (Commit4.5).
- Supplementary video 4: Modelled ice-thickness evolution 2101–2300 (Commit8.5).

## 810 Appendix A: Additional figures

815

820

Author contributions. The study was conceptualised and formulated by HÅ, KHS, TVS and TD. HÅ carried out pre-processing of most input data, conducted dynamical model simulations, analysed model results, wrote most of the manuscript, and made all figures except Fig. 1 (made by BAR) and Figs. 4, 5, 12, A5 (made by KHS). Model development and simulations of surface mass balance were done by KHS, who also pre-processed and analysed climate model data, and wrote substantial parts of the manuscript. HÅ and KHS jointly coded the coupling for dynamics and surface mass balance. TVS and TD contributed with advice on methodology, model strategy, visualisation, and comments on scientific writing. MKG and LMA prepared the ice-thickness data. LMA also provided guidance with data on glacier outlines and front positions. BAR and LMA prepared historical data for ice geometry. TS contributed with processing and advice on ice-velocity data. JCY contributed with funding acquisition, project management, and writing/editing. All authors contributed to writing and revising the manuscript, and approved the final version for submission.

Competing interests. The authors declare that they have no conflict of interest.

Figure A1. Model mesh on underlying bed topography, as described in Sect. 4.

Acknowledgements. This work is a contribution to the JOSTICE project funded by the Research Council of Norway (RCN grant #302458) and ERC-2022-ADG grant agreement No 01096057 GLACMASS from the European Research Council. TS was funded under MASSIVE (RCN #315971). We would like to thank Wai Kwok Wong at the Norwegian Water Resources and Energy Directorate (NVE) for providing input on the selection of climate model and emission scenarios used in this study. The simulations in the study were performed thanks to computing and storage resources provided by the IT section of the Department of Geosciences at the University of Oslo. We would like to thank the editor Horst Machguth, reviewer Ward van Pelt, and one anonymous reviewer for their constructive comments, which helped to improve the paper.

**Figure A2.** Simulated historical change (a–c) and future (d–f) evolution with experiment Future4.5-ECC (cf. Table 2), using a 20 m ice thickness threshold (cf. Fig. 2, which uses a 10 m threshold). Recent historical periods of little change (gray), ice-cap growth (blue) and decay (red) are shaded in (a–c). In (c), historical SMB with evolving model area is shown (SMB<sub>transArea</sub>), but without elevation changes, since these make little difference (Sect. 5.2). Shown in (f) is SMB with transiently evolving geometry (SMB<sub>transGeom</sub>), including the SMB–elevation feedback and area changes.

**Figure A3.** Model evolution of (a) SMB<sub>static</sub>, (b) ice volume, and (c) area in experiments aiming to grow Jostedalsbreen from ice-free conditions. See Table 2 for experiment details.

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

**Figure A4.** Future simulated mean annual SMB<sub>transGeom</sub>, ELA and accumulation-area ratio (AAR) under (a–c) RCP4.5 and (d–f) RCP8.5 emission pathways, using the climate model ECEARTH/CCLM. Yearly AAR and 10-year running means are shown in (b) and (e). Present-day glacier outlines (2019) are shown with black lines (Andreassen et al., 2022) on the SMB maps.

Andreassen, L. M., Nagy, T., Kjøllmoen, B., and Leigh, J. R.: An inventory of Norway's glaciers and ice-marginal lakes from 2018–19 Sentinel-2 data, Journal of Glaciology, pp. 1–22, https://doi.org/10.1017/jog.2022.20, 2022.

Andreassen, L. M., Robson, B. A., Sjursen, K. H., Elvehøy, H., Kjøllmoen, B., and Carrivick, J. L.: Spatio-temporal variability in geometry and geodetic mass balance of Jostedalsbreen ice cap, Norway, Annals of Glaciology, pp. 1–18, https://doi.org/10.1017/aog.2023.70, publisher: Cambridge University Press, 2023.

ArcGIS Development Team: Esri Inc.: ArcGIS Pro (Version 3.4), https://www.esri.com/en-us/arcgis/products/arcgis-pro/overview (last access: 25 August 2025), 2025.

Aschwanden, A., Aðalgeirsdóttir, G., and Khroulev, C.: Hindcasting to measure ice sheet model sensitivity to initial states, The Cryosphere, 7, 1083–1093, https://doi.org/10.5194/tc-7-1083-2013, publisher: Copernicus GmbH, 2013.

Aðalgeirsdóttir, G., Gudmundsson, G. H., and Björnsson, H.: Volume sensitivity of Vatnajökull Ice Cap, Iceland, to perturbations in equilibrium line altitude, Journal of Geophysical Research: Earth Surface, 110, n/a–n/a, https://doi.org/10.1029/2005JF000289, 2005.

Ađalgeirsdóttir, G., Aschwanden, A., Khroulev, C., Boberg, F., Mottram, R., Lucas-Picher, P., and Christensen, J.: Role of model initialization for projections of 21st-century Greenland ice sheet mass loss, Journal of Glaciology, 60, 782–794, https://doi.org/10.3189/2014JoG13J202, 2014.

**Figure A5.** Future changes in (a, b) mean temperature and (c, d) total precipitation in (a, c) winter (Oct–Apr) and (b, d) summer (May–Sep) months for the ensemble of climate model forcing.

- Bickerton, R. W. and Matthews, J. A.: 'Little ice age' variations of outlet glaciers from the jostedalsbreen ice-cap, Southern Norway: A regional lichenometric-dating study of ice-marginal moraine sequences and their climatic significance, Journal of Quaternary Science, 8, 45–66, https://doi.org/10.1002/jqs.3390080105, \_eprint: https://onlinelibrary.wiley.com/doi/pdf/10.1002/jqs.3390080105, 1993.
- Blatter, H.: Velocity and stress fields in grounded glaciers: a simple algorithm for including deviatoric stress gradients, Journal of Glaciology, 41, 333–344, https://doi.org/10.3189/S002214300001621X, 1995.
  - Box, J. E., Hubbard, A., Bahr, D. B., Colgan, W. T., Fettweis, X., Mankoff, K. D., Wehrlé, A., Noël, B., van den Broeke, M. R., Wouters, B., Bjørk, A. A., and Fausto, R. S.: Greenland ice sheet climate disequilibrium and committed sea-level rise, Nature Climate Change, 12, 808–813, https://doi.org/10.1038/s41558-022-01441-2, publisher: Nature Publishing Group, 2022.
- Brondex, J., Gillet-Chaulet, F., and Gagliardini, O.: Sensitivity of centennial mass loss projections of the Amundsen basin to the friction law, The Cryosphere, 13, 177–195, https://doi.org/10.5194/tc-13-177-2019, 2019.
  - Budd, W. F., Keage, P. L., and Blundy, N. A.: Empirical Studies of Ice Sliding, Journal of Glaciology, 23, 157–170, https://doi.org/10.3189/S0022143000029804, 1979.
- Carrivick, J. L., Andreassen, L. M., Nesje, A., and Yde, J. C.: A reconstruction of Jostedalsbreen during the Little Ice Age and geometric changes to outlet glaciers since then, Quaternary Science Reviews, 284, 107 501, https://doi.org/10.1016/j.quascirev.2022.107501, 2022.

**Table A1.** Overview of data sources used in this study.

| Data type               | Aquisition method               | Time period | Resolution (m) | Usage       | Reference                     |
|-------------------------|---------------------------------|-------------|----------------|-------------|-------------------------------|
| Subglacial topography   | ice radar                       | 2019-2023   | 10             | Input       | Gillespie et al. (2024)       |
| Land-surface topography | aerial imagery                  | 2019        | 10             | Input       | Andreassen et al. (2023)      |
| Ice-surface topography  | aerial imagery                  | 1966        | 10             | Input       | Andreassen et al. (2023)      |
| Ice-surface topography  | topographic map                 | 1966        | 50             | Input       | Paul et al. (2011)            |
| Ice-surface topography  | aerial imagery                  | 2019        | 10             | Validation  | Andreassen et al. (2023)      |
| Ice-surface velocity    | radar+optical satellite imagery | 2017-2018   | 50             | Calibration | Millan et al. (2022)          |
| Ice-surface velocity    | radar satellite imagery         | 1996        | 20             | Calibration | this study (Sect. 3.2)        |
| Ice-surface velocity    | optical satellite imagery       | 2015-2018   | 80             | Validation  | Nagy and Andreassen (2019)    |
| Glacier outlines        | optical satellite imagery       | 1966, 2019  | 10             | Validation  | Winsvold et al. (2014)        |
|                         |                                 |             |                |             | Andreassen et al. (2022)      |
| Glacier front positions | in-situ                         | various     | 1              | Validation  | Andreassen and Elvehøy (2021) |
|                         |                                 |             |                |             | Andreassen et al. (2023)      |
|                         |                                 |             |                |             | Kjøllmoen et al. (2024)       |

- Compagno, L., Zekollari, H., Huss, M., and Farinotti, D.: Limited impact of climate forcing products on future glacier evolution in Scandinavia and Iceland, Journal of Glaciology, 67, 727–743, https://doi.org/10.1017/jog.2021.24, 2021.
- Cuffey, K. M. and Paterson, W. S. B.: The Physics of Glaciers, Academic Press, 2010.

- Dannevig, H. and Rusdal, T.: Caring for melting glaciers, Tourism Geographies, 25, 1679–1695, 870 https://doi.org/10.1080/14616688.2023.2278762, 2023.
  - Davies, B., McNabb, R., Bendle, J., Carrivick, J., Ely, J., Holt, T., Markle, B., McNeil, C., Nicholson, L., and Pelto, M.: Accelerating glacier volume loss on Juneau Icefield driven by hypsometry and melt-accelerating feedbacks, Nature Communications, 15, 5099, https://doi.org/10.1038/s41467-024-49269-y, publisher: Nature Publishing Group, 2024.
  - de Fleurian, B., Davy, R., and Langebroek, P. M.: Impact of runoff temporal distribution on ice dynamics, The Cryosphere, 16, 2265–2283, https://doi.org/10.5194/tc-16-2265-2022, publisher: Copernicus GmbH, 2022.
  - Ekblom Johansson, F., Bakke, J., Støren, E. N., Gillespie, M. K., and Laumann, T.: Mapping of the Subglacial Topography of Folgefonna Ice Cap in Western Norway—Consequences for Ice Retreat Patterns and Hydrological Changes, Frontiers in Earth Science, 10, https://www.frontiersin.org/article/10.3389/feart.2022.886361, 2022.
- Elsberg, D. H., Harrison, W. D., Echelmeyer, K. A., and Krimmel, R. M.: Quantifying the effects of climate and surface change on glacier mass balance, Journal of Glaciology, 47, 649–658, https://doi.org/10.3189/172756501781831783, 2001.
  - Erikstad, L. and Sollid, J. L.: Neoglaciation in South Norway using lichenometric methods, Norsk Geografisk Tidsskrift Norwegian Journal of Geography, 40, 85–105, https://doi.org/10.1080/00291958608552159, 1986.
  - Farinotti, D., Huss, M., Fürst, J. J., Landmann, J., Machguth, H., Maussion, F., and Pandit, A.: A consensus estimate for the ice thickness distribution of all glaciers on Earth, Nature Geoscience, 12, 168–173, https://doi.org/10.1038/s41561-019-0300-3, 2019.

- Frank, T. and van Pelt, W. J. J.: Ice volume and thickness of all Scandinavian glaciers and ice caps, Journal of Glaciology, pp. 1–14, https://doi.org/10.1017/jog.2024.25, 2024.
  - Friedl, P., Seehaus, T., and Braun, M.: Global time series and temporal mosaics of glacier surface velocities derived from Sentinel-1 data, Earth System Science Data, 13, 4653–4675, https://doi.org/10.5194/essd-13-4653-2021, publisher: Copernicus GmbH, 2021.
  - Gabbi, J., Carenzo, M., Pellicciotti, F., Bauder, A., and Funk, M.: A comparison of empirical and physically based glacier surface melt models for long-term simulations of glacier response, Journal of Glaciology, 60, 1140–1154, https://doi.org/10.3189/2014JoG14J011, 2014.

- GAMMA AG: GAMMA Remote Sensing Research and Consulting AG (Version 20160625), https://https://www.gamma-rs.ch/gamma-software/gamma-software (last access: 04 September 2025), 2016.
- Garbe, J., Albrecht, T., Levermann, A., Donges, J. F., and Winkelmann, R.: The hysteresis of the Antarctic Ice Sheet, Nature, 585, 538–544, https://doi.org/10.1038/s41586-020-2727-5, publisher: Nature Publishing Group, 2020.
- Giesen, R. H. and Oerlemans, J.: Response of the ice cap Hardangerjøkulen in southern Norway to the 20th and 21st century climates, The Cryosphere, 4, 191–213, https://doi.org/10.5194/tc-4-191-2010, 2010.
  - Gilbert, A., Flowers, G. E., Miller, G. H., Rabus, B. T., Van Wychen, W., Gardner, A. S., and Copland, L.: Sensitivity of Barnes Ice Cap, Baffin Island, Canada, to climate state and internal dynamics: Barnes Ice Cap Stability, Journal of Geophysical Research: Earth Surface, 121, 1516–1539, https://doi.org/10.1002/2016JF003839, 2016.
- Gillespie, M. K., Andreassen, L. M., Huss, M., de Villiers, S., Sjursen, K. H., Aasen, J., Bakke, J., Cederstrøm, J. M., Elvehøy, H., Kjøllmoen, B., Loe, E., Meland, M., Melvold, K., Nerhus, S. D., Røthe, T. O., Støren, E. W. N., Øst, K., and Yde, J. C.: Ice thickness and bed topography of Jostedalsbreen ice cap, Norway, Earth System Science Data, 16, 5799–5825, https://doi.org/10.5194/essd-16-5799-2024, publisher: Copernicus GmbH, 2024.
- Gillet-Chaulet, F., Gagliardini, O., Seddik, H., Nodet, M., Durand, G., Ritz, C., Zwinger, T., Greve, R., and Vaughan, D. G.: Greenland ice sheet contribution to sea-level rise from a new-generation ice-sheet model, The Cryosphere, 6, 1561–1576, https://doi.org/10.5194/tc-6-1561-2012, 2012.
  - Gjerde, M., Hoel, O. L., and Nesje, A.: The 'Little Ice Age' advance of Nigardsbreen, Norway: A cross-disciplinary revision of the chronological framework, The Holocene, p. 09596836231185830, https://doi.org/10.1177/09596836231185830, publisher: SAGE Publications Ltd, 2023.
- Goldberg, D. N., Heimbach, P., Joughin, I., and Smith, B.: Committed retreat of Smith, Pope, and Kohler Glaciers over the next 30 years inferred by transient model calibration, The Cryosphere, 9, 2429–2446, https://doi.org/10.5194/tc-9-2429-2015, 2015.
  - Gulbrandsen, T.: WF-1833 SOGNDAL-JOSTEDALSBREEN-GEIRANGER 1966. Rapport bildematcing av—Historiske ortofoto, Tech. rep., Hexagon, Fredrikstad, Norway, 2022.
- Hacker, B. R., Andersen, T. B., Johnston, S., Kylander-Clark, A. R. C., Peterman, E. M., Walsh, E. O., and Young, D.: High-temperature
   deformation during continental-margin subduction & exhumation: The ultrahigh-pressure Western Gneiss Region of Norway, Tectonophysics, 480, 149–171, https://doi.org/10.1016/j.tecto.2009.08.012, 2010.
  - Hanssen-Bauer, I., Førland, E., Haddeland, I., Hisdal, H., Mayer, S., Nesje, A., Nilsen, J., Sandven, S. A., Sandø, A., Sorteberg, A., and Ådlandsvi, B.: Climate in Norway 2100, Norwegian Center for Climate Services (NCCS) Report 1/2017, Norwegian Environment Agency (Miljødirektoratet), 2017.
- Harrison, W. D., Elsberg, D. H., Echelmeyer, K. A., and Krimmel, R. M.: On the characterization of glacier response by a single time-scale, Journal of Glaciology, 47, 659–664, https://doi.org/10.3189/172756501781831837, 2001.

- Hugonnet, R., McNabb, R., Berthier, E., Menounos, B., Nuth, C., Girod, L., Farinotti, D., Huss, M., Dussaillant, I., Brun, F., and Kääb, A.: Accelerated global glacier mass loss in the early twenty-first century, Nature, 592, 726–731, https://doi.org/10.1038/s41586-021-03436-z, 2021.
- Huss, M. and Hock, R.: A new model for global glacier change and sea-level rise, Frontiers in Earth Sciences, 3, https://doi.org/10.3389/feart.2015.00054, 2015.
  - Huss, M. and Hock, R.: Global-scale hydrological response to future glacier mass loss, Nature Climate Change, 8, 135–140, https://doi.org/10.1038/s41558-017-0049-x, 2018.
- Huss, M., Hock, R., Bauder, A., and Funk, M.: Conventional versus reference-surface mass balance, Journal of Glaciology, 58, 278–286, https://doi.org/10.3189/2012JoG11J216, 2012.
  - Huss, M., Bookhagen, B., Huggel, C., Jacobsen, D., Bradley, R., Clague, J., Vuille, M., Buytaert, W., Cayan, D., Greenwood, G., Mark, B., Milner, A., Weingartner, R., and Winder, M.: Toward mountains without permanent snow and ice, Earth's Future, 5, 418–435, https://doi.org/10.1002/2016EF000514, 2017.
  - Hutchinson, M. F., Xu, T., Stein, J. A., et al.: Recent progress in the ANUDEM elevation gridding procedure, Geomorphometry, 2011, 19–22, 2011.

- Hutter, K.: Theoretical glaciology: material science of ice and the mechanics of glaciers and ice sheets, vol. 1, Springer, https://doi.org/10.1007/978-94-015-1167-4, 1983.
- Iken, A.: The Effect of the Subglacial Water Pressure on the Sliding Velocity of a Glacier in an Idealized Numerical Model, Journal of Glaciology, 27, 407–421, https://doi.org/10.3189/S0022143000011448, 1981.
- Ismail, M. F., Bogacki, W., Disse, M., Schäfer, M., and Kirschbauer, L.: Estimating degree-day factors of snow based on energy flux components, The Cryosphere, 17, 211–231, https://doi.org/10.5194/tc-17-211-2023, 2023.
  - Jacob, D., Petersen, J., Eggert, B., Alias, A., Christensen, O., Bouwer, L., Braun, A., Colette, A., Déqué, M., Georgievski, G., Georgopoulou, E., Gobiet, A., Menut, L., Nikulin, G., Haensler, A., Hempelmann, N., Jones, C., Keuler, K., Kovats, S., and Yiou, P.: EURO-CORDEX: New high-resolution climate change projections for European impact research, Regional Environmental Change, 14, https://doi.org/10.1007/s10113-013-0499-2, 2014.
  - Jostedalsbreen nasjonalparkstyre: Besøksstrategi Jostedalsbreen nasjonalpark 2021-2027, Tech. rep., https://www.nasjonalparkstyre.no/uploads/files\_jostedalsbreen/Besoksstrategi-Jostedalsbreen-2021-2027-godkjent.pdf, 2021.
  - Jouvet, G. and Cordonnier, G.: Ice-flow model emulator based on physics-informed deep learning, Journal of Glaciology, pp. 1–15, https://doi.org/10.1017/jog.2023.73, publisher: Cambridge University Press, 2023.
- Jouvet, G., Huss, M., Blatter, H., Picasso, M., and Rappaz, J.: Numerical simulation of Rhonegletscher from 1874 to 2100, Journal of Computational Physics, 228, 6426–6439, https://doi.org/10.1016/j.jcp.2009.05.033, 2009.
  - Jouvet, G., Huss, M., Funk, M., and Blatter, H.: Modelling the retreat of Grosser Aletschgletscher, Switzerland, in a changing climate, Journal of Glaciology, 57, 1033–1045, https://doi.org/10.3189/002214311798843359, 2011.
- Jouvet, G., Cordonnier, G., Kim, B., Lüthi, M., Vieli, A., and Aschwanden, A.: Deep learning speeds up ice flow modelling by several orders of magnitude, Journal of Glaciology, pp. 1–14, https://doi.org/10.1017/jog.2021.120, 2021.
  - Jóhannesson, T., Raymond, C., and Waddington, E.: Time–Scale for Adjustment of Glaciers to Changes in Mass Balance, Journal of Glaciology, 35, 355–369, https://doi.org/10.3189/S002214300000928X, 1989.
  - Jóhannesson, T., Sigurdsson, O., Laumann, T., and Kennett, M.: Degree-day glacier mass-balance modelling with applications to glaciers in Iceland, Norway and Greenland, Journal of Glaciology, 41, 345–358, https://doi.org/10.3189/S0022143000016221, 1995.

- Ketzler, G., Römer, W., and Beylich, A. A.: The Climate of Norway, pp. 7–29, Springer International Publishing, Cham, https://doi.org/10.1007/978-3-030-52563-7\_2, 2021.
  - Kjøllmoen, B., Andreassen, L. M., and Elvehøy, H.: Glaciological investigations in Norway 2023, NVE Rapport 22-2024, Tech. Rep. 22-2024, http://publikasjoner.nve.no/rapport/2024/rapport2024\_22.pdf, 2024.
- Larour, E., Seroussi, H., Morlighem, M., and Rignot, E.: Continental scale, high order, high spatial resolution, ice sheet modeling using
  the Ice Sheet System Model (ISSM): ICE SHEET SYSTEM MODEL, Journal of Geophysical Research: Earth Surface, 117, n/a–n/a,
  https://doi.org/10.1029/2011JF002140, 2012.
  - Laumann, T. and Nesje, A.: The impact of climate change on future frontal variations of Briksdalsbreen, western Norway, Journal of Glaciology, 55, 789–796, https://doi.org/10.3189/002214309790152366, 2009.
- Laumann, T. and Nesje, A.: Spørteggbreen, western Norway, in the past, present and future: Simulations with a two-dimensional dynamical glacier model, The Holocene, 24, 842–852, https://doi.org/10.1177/0959683614530446, 2014.
  - Le Meur, E. and Vincent, C.: A two-dimensional shallow ice-flow model of Glacier de Saint-Sorlin, France, Journal of Glaciology, 49, 527–538, https://doi.org/10.3189/172756503781830421, 2003.
  - Le Meur, E., Gagliardini, O., Zwinger, T., and Ruokolainen, J.: Glacier flow modelling: a comparison of the Shallow Ice Approximation and the full-Stokes solution, Comptes Rendus. Physique, 5, 709–722, https://doi.org/10.1016/j.crhy.2004.10.001, 2004.
- Leysinger Vieli, G. J.-M. C. and Gudmundsson, G. H.: On estimating length fluctuations of glaciers caused by changes in climatic forcing, Journal of Geophysical Research: Earth Surface, 109, https://doi.org/10.1029/2003JF000027, 2004.
  - Lussana, C.: seNorge observational gridded datasets, seNorge\_2018, versions 21.09 and 21.10, METreport 07/2021, The Norwegian Meteorological Institute (MET Norway), Oslo, Norway, 2021.
- Lussana, C., Tveito, O., Dobler, A., and Tunheim, K.: seNorge\_2018, daily precipitation, and temperature datasets over Norway, Earth System Science Data, 11, 1531–1551, https://doi.org/10.5194/essd-11-1531-2019, 2019.
  - Marshall, S. J.: Simulation of Vatnajökull ice cap dynamics, Journal of Geophysical Research, 110, F03 009, https://doi.org/10.1029/2004JF000262, 2005.
  - Marzeion, B., Jarosch, A. H., and Hofer, M.: Past and future sea-level change from the surface mass balance of glaciers, The Cryosphere, 6, 1295–1322, https://doi.org/10.5194/tc-6-1295-2012, publisher: Copernicus GmbH, 2012.
- Maussion, F., Butenko, A., Champollion, N., Dusch, M., Eis, J., Fourteau, K., Gregor, P., Jarosch, A. H., Landmann, J., Oesterle, F., Recinos, B., Rothenpieler, T., Vlug, A., Wild, C. T., and Marzeion, B.: The Open Global Glacier Model (OGGM) v1.1, Geoscientific Model Development, 12, 909–931, https://doi.org/10.5194/gmd-12-909-2019, publisher: Copernicus GmbH, 2019.
  - Meinshausen, M., Nicholls, Z. R. J., Lewis, J., Gidden, M. J., Vogel, E., Freund, M., Beyerle, U., Gessner, C., Nauels, A., Bauer, N., Canadell, J. G., Daniel, J. S., John, A., Krummel, P. B., Luderer, G., Meinshausen, N., Montzka, S. A., Rayner, P. J., Reimann, S., Smith, S. J., van den Berg, M., Velders, G. J. M., Vollmer, M. K., and Wang, R. H. J.: The shared socio-economic pathway (SSP) greenhouse gas concentrations and their extensions to 2500, Geoscientific Model Development, 13, 3571–3605, https://doi.org/10.5194/gmd-13-3571-2020, publisher: Copernicus GmbH, 2020.

- Millan, R., Mouginot, J., Rabatel, A., and Morlighem, M.: Ice velocity and thickness of the world's glaciers, Nature Geoscience, 15, 124–129, https://doi.org/10.1038/s41561-021-00885-z, number: 2 Publisher: Nature Publishing Group, 2022.
- Milner, A. M., Khamis, K., Battin, T. J., Brittain, J. E., Barrand, N. E., Füreder, L., Cauvy-Fraunié, S., Gíslason, G. M., Jacobsen, D., Hannah, D. M., Hodson, A. J., Hood, E., Lencioni, V., Ólafsson, J. S., Robinson, C. T., Tranter, M., and Brown, L. E.:

- Glacier shrinkage driving global changes in downstream systems, Proceedings of the National Academy of Sciences, 114, 9770–9778, https://doi.org/10.1073/pnas.1619807114, publisher: Proceedings of the National Academy of Sciences, 2017.
- Mohr, M.: New Routines for Gridding of Temperature and Precipitation Observations for "seNorge. no", Tech. Rep. 08/2008, Norwegian Meteorological Institute, Oslo, Norway, 2008.
  - Morlighem, M., Rignot, E., Seroussi, H., Larour, E., Ben Dhia, H., and Aubry, D.: Spatial patterns of basal drag inferred using control methods from a full-Stokes and simpler models for Pine Island Glacier, West Antarctica, Geophysical Research Letters, 37, n/a–n/a, https://doi.org/10.1029/2010GL043853, 2010.
  - Nagy, T. and Andreassen, L.: Glacier surface velocity mapping with Sentinel-2 imagery in Norway. NVE Report 37, Tech. rep., 2019.
- Nesje, A. and Matthews, J. A.: The Briksdalsbre Event: A winter precipitation-induced decadal-scale glacial advance in southern Norway in the 1990s and its implications, The Holocene, 22, 249–261, https://doi.org/10.1177/0959683611414938, 2012.
  - Nesje, A., Lie, O., and Dahl, S. O.: Is the North Atlantic Oscillation reflected in Scandinavian glacier mass balance records?, Journal of Quaternary Science, 15, 587–601, https://doi.org/10.1002/1099-1417(200009)15:6<587::AID-JQS533>3.0.CO;2-2, 2000.
- Noël, B., van de Berg, W. J., Machguth, H., Lhermitte, S., Howat, I., Fettweis, X., and van den Broeke, M. R.: A daily, 1 km resolution data set of downscaled Greenland ice sheet surface mass balance (1958–2015), The Cryosphere, 10, 2361–2377, https://doi.org/10.5194/tc-10-2361-2016, 2016.
  - NVE: Climate indicator products, https://glacier.nve.no/glacier/viewer/ci/en/, Last accessed on 28 Aug 2025, 2025.
  - Oerlemans, J.: Some basic experiments with a vertically-integrated ice sheet model, Tellus, 33, 1–11, https://doi.org/10.1111/j.2153-3490.1981.tb01726.x, \_eprint: https://onlinelibrary.wiley.com/doi/pdf/10.1111/j.2153-3490.1981.tb01726.x, 1981.
- Oerlemans, J.: A flowline model for Nigardsbreen, Norway: projection of future glacier length based on dynamic calibration with the historic record, Annals of Glaciology, 24, 382–389, https://doi.org/10.3189/S0260305500012489, 1997.
  - Pattyn, F.: A new three-dimensional higher-order thermomechanical ice sheet model: Basic sensitivity, ice stream development, and ice flow across subglacial lakes, Journal of Geophysical Research: Solid Earth, 108, https://doi.org/10.1029/2002JB002329, \_eprint: https://onlinelibrary.wiley.com/doi/pdf/10.1029/2002JB002329, 2003.
- Paul, F., Andreassen, L. M., and Winsvold, S. H.: A new glacier inventory for the Jostedalsbreen region, Norway, from Landsat TM scenes of 2006 and changes since 1966, Annals of Glaciology, 52, 153–162, https://doi.org/10.3189/172756411799096169, 2011.
  - Pollard, D. and DeConto, R. M.: A simple inverse method for the distribution of basal sliding coefficients under ice sheets, applied to Antarctica, The Cryosphere, 6, 953–971, https://doi.org/10.5194/tc-6-953-2012, 2012.
  - QGIS Development Team: QGIS Geographic Information System, QGIS Association, https://www.qgis.org, 2024.

- Réveillet, M., Six, D., Vincent, C., Rabatel, A., Dumont, M., Lafaysse, M., Morin, S., Vionnet, V., and Litt, M.: Relative performance of empirical and physical models in assessing the seasonal and annual glacier surface mass balance of Saint-Sorlin Glacier (French Alps), The Cryosphere, 12, 1367–1386, https://doi.org/10.5194/tc-12-1367-2018, 2018.
  - Rounce, D. R., Hock, R., Maussion, F., Hugonnet, R., Kochtitzky, W., Huss, M., Berthier, E., Brinkerhoff, D., Compagno, L., Copland, L., Farinotti, D., Menounos, B., and McNabb, R. W.: Global glacier change in the 21st century: Every increase in temperature matters, Science, 379, 78–83, https://doi.org/10.1126/science.abo1324, publisher: American Association for the Advancement of Science, 2023.
  - Santos, T. D. D., Morlighem, M., Simões, J. C., and Devloo, P. R. B.: Sensitivity analysis of a King George Island outlet glacier, South Shetlands, Antarctica, Anais da Academia Brasileira de Ciências, 95, e20210 560, https://doi.org/10.1590/0001-3765202320210560, publisher: Academia Brasileira de Ciências, 2023.

- Schaffer, N., Copland, L., Zdanowicz, C., and Hock, R.: Modeling the surface mass balance of Penny Ice Cap, Baffin Island, 1959–2099,

  Annals of Glaciology, 64, 330–342, https://doi.org/10.1017/aog.2023.68, 2023.
  - Schmidt, L. S., Aðalgeirsdóttir, G., Pálsson, F., Langen, P. L., Guðmundsson, S., and Björnsson, H.: Dynamic simulations of Vatnajökull ice cap from 1980 to 2300, Journal of Glaciology, 66, 97–112, https://doi.org/10.1017/jog.2019.90, 2020.
- Seier, G., Abermann, J., Andreassen, L. M., Carrivick, J. L., Kielland, P. H., Löffler, K., Nesje, A., Robson, B. A., Røthe, T. O., Scheiber, T., Winkler, S., and Yde, J. C.: Glacier thinning, recession and advance, and the associated evolution of a glacial lake between 1966 and 2021 at Austerdalsbreen, western Norway, Land Degradation & Development, 35, 394–414, https://doi.org/10.1002/ldr.4923, \_eprint: https://onlinelibrary.wiley.com/doi/pdf/10.1002/ldr.4923, 2024.
  - Sjursen, K. H., Dunse, T., Tambue, A., Schuler, T. V., and Andreassen, L. M.: Bayesian parameter estimation in glacier mass-balance modelling using observations with distinct temporal resolutions and uncertainties, Journal of Glaciology, p. 1–20, https://doi.org/10.1017/jog.2023.62, 2023.
- Sjursen, K. H., Dunse, T., Schuler, T. V., Andreassen, L. M., and Åkesson, H.: Spatiotemporal mass balance variability of Jostedalsbreen Ice Cap, Norway, revealed by a temperature-index model using Bayesian inference, Annals of Glaciology, 66, 1–18, https://doi.org/10.1017/aog.2024.41, 2025.
  - Troch, M., Åkesson, H., Cuzzone, J. K., and Bertrand, S.: Precipitation drives western Patagonian glacier variability and may curb future ice mass loss, Scientific Reports, 14, 26744, https://doi.org/10.1038/s41598-024-77486-4, publisher: Nature Publishing Group, 2024.
- Tvede, A.: Folgefonni, en glasiologisk avviker, Naturen, 97, 11–16, 1973.
  - Tvede, A. and Liestøl, O.: Blomsterskardbreen, Folgefonni, mass balance and recent fluctuations, Nor. Polarinst. Arbok, pp. 225–233, 1976.
  - Tvede, A. M.: Floods Caused by a Glacier-Dammed Lake at the Folgefonni Ice Cap, Norway, Annals of Glaciology, 13, 262–264, https://doi.org/10.3189/S0260305500008016, 1989.
- United Nations Environment Programme: Emissions Gap Report 2024: No more hot air ... please! With a massive gap between rhetoric and reality, countries draft new climate commitments., Tech. rep., Nairobi, https://doi.org/10.59117/20.500.11822/46404, section: publications, 2024.
  - van Pelt, W. J. J., Schuler, T. V., Pohjola, V. A., and Pettersson, R.: Accelerating future mass loss of Svalbard glaciers from a multi-model ensemble, Journal of Glaciology, 67, 485–499, https://doi.org/10.1017/jog.2021.2, 2021.
- Van Tricht, L. and Huybrechts, P.: Modelling the historical and future evolution of six ice masses in the Tien Shan, Central Asia, using a 3D ice-flow model, The Cryosphere, 17, 4463–4485, https://doi.org/10.5194/tc-17-4463-2023, publisher: Copernicus GmbH, 2023.
  - Verfaillie, D., Charton, J., Schimmelpfennig, I., Stroebele, Z., Jomelli, V., Bétard, F., Favier, V., Cavero, J., Berthier, E., Goosse, H., Rinterknecht, V., Legentil, C., Charrassin, R., Aumaître, G., Bourlès, D. L., and Keddadouche, K.: Evolution of the Cook Ice Cap (Kerguelen Islands) between the last centuries and 2100 ce based on cosmogenic dating and glacio-climatic modelling, Antarctic Science, 33, 301–317, https://doi.org/10.1017/S0954102021000080, 2021.
- Wangensteen, B., Tønsberg, O. M., Kääb, A., Eiken, T., and Hagen, J. O.: Surface Elevation Change and High Resolution Surface Velocities for Advancing Outlets of Jostedalsbreen, Geografiska Annaler: Series A, Physical Geography, 88, 55–74, https://doi.org/10.1111/j.0435-3676.2006.00283.x, \_eprint: https://onlinelibrary.wiley.com/doi/pdf/10.1111/j.0435-3676.2006.00283.x, 2006.
- Welty, E., Zemp, M., Navarro, F., Huss, M., Fürst, J. J., Gärtner-Roer, I., Landmann, J., Machguth, H., Naegeli, K., Andreassen, L. M., Farinotti, D., Li, H., and Contributors, G.: Worldwide version-controlled database of glacier thickness observations, Earth System Science Data, 12, 3039–3055, https://doi.org/10.5194/essd-12-3039-2020, publisher: Copernicus GmbH, 2020.

- Winsvold, S. H., Andreassen, L. M., and Kienholz, C.: Glacier area and length changes in Norway from repeat inventories, The Cryosphere, 8, 1885–1903, 2014.
- Wong, W. K., Haddeland, I., Lawrence, D., and Beldring, S.: Gridded 1 x 1 km climate and hydrological projections for Norway, NVE report 59/2016, Norwegian Water Resources and Energy Directorate (NVE), Oslo, Norway, 2016.
- Zekollari, H., Huybrechts, P., Noël, B., van de Berg, W. J., and van den Broeke, M. R.: Sensitivity, stability and future evolution of the world's northernmost ice cap, Hans Tausen Iskappe (Greenland), The Cryosphere, 11, 805–825, https://doi.org/10.5194/tc-11-805-2017, publisher: Copernicus GmbH, 2017.

- Zekollari, H., Huss, M., Farinotti, D., and Lhermitte, S.: Ice-Dynamical Glacier Evolution Modeling—A Review, Reviews of Geophysics, 60, e2021RG000754, https://doi.org/10.1029/2021RG000754, \_eprint: https://onlinelibrary.wiley.com/doi/pdf/10.1029/2021RG000754, 2022.
- Zemp, M. and Haeberli, W.: Glaciers and ice caps. Part I: Global overview and outlook. Part II: Glacier changes around the world, UNEP, Nairobi, https://doi.org/10.5167/uzh-40427, 2007.
- Ziemen, F. A., Hock, R., Aschwanden, A., Khroulev, C., Kienholz, C., Melkonian, A., and Zhang, J.: Modeling the evolution of the Juneau Icefield between 1971 and 2100 using the Parallel Ice Sheet Model (PISM), Journal of Glaciology, 62, 199–214, https://doi.org/10.1017/jog.2016.13, 2016.
- Åkesson, H., Nisancioglu, K. H., Giesen, R. H., and Morlighem, M.: Simulating the evolution of Hardangerjøkulen ice cap in southern Norway since the mid-Holocene and its sensitivity to climate change, The Cryosphere, 11, 281–302, https://doi.org/10.5194/tc-11-281-2017, 2017.
- Åkesson, H., Morlighem, M., Nisancioglu, K. H., Svendsen, J. I., and Mangerud, J.: Atmosphere-driven ice sheet mass loss paced by topography: Insights from modelling the south-western Scandinavian Ice Sheet, Quaternary Science Reviews, 195, 32–47, https://doi.org/10.1016/j.quascirev.2018.07.004, 2018.
  - Åkesson, H., Morlighem, M., O'Regan, M., and Jakobsson, M.: Future Projections of Petermann Glacier Under Ocean Warming Depend Strongly on Friction Law, Journal of Geophysical Research: Earth Surface, 126, https://doi.org/10.1029/2020JF005921, 2021.