# Peer review of "Recent history and future demise of Jostedalsbreen, the largest ice cap in mainland Europe"

_EGUsphere, 2025_

## Author Comment (AC2)

**Authors Response to Reviews - Åkesson et al. egusphere-2025-467**

September 4, 2025

Dear editor and reviewers,
We would like to thank the editor and the two reviewers for their positive and constructive comments. We detail our responses and changes to the manuscript below.
Sincerely,
the authors

**Editor's comments**

**Comments from rapid access review**

I have no suggestions for immediate changes. However, there are a few points that could be addressed during the review process. The terminology around capping emissions is not always clear to me. For example, it says on Line 488 "if greenhouse-gas emissions were capped at present-day levels". I am not sure what would be the optimal terminology, but this and other similar statements could be misunderstood in the sense of "if annual amounts of greenhouse- gas emissions remain at today's level". However, I believe what is actually meant is "if the atmospheric concentration of greenhouse-gases would be stabilized at today's level", which would require a very massive reduction in annual emissions. Furthermore, is it correct that stabilizing greenhouse-gas emissions at the current level would lead to a rather immediate stabilization of the climate (see my next comment)? The manuscript mentions e.g. "if future greenhouse-gas emissions are rolled back immediately at the year 2100" (line 500) and thereby refers to "Reverse experiments explore the potential to backpedal the modelled 21st-century evolution of the ice cap (Reverse4.5 and 8.5). To this end, mean annual SMB of 2001–2020 is used as forcing." (line 363). However, I do not think that rolling back (what exactly is meant; rolling back to zero?) of greenhouse-gas emissions in the year 2100 will result in an immediate change of the climate back to the conditions of the early 21st century. There is no need to change the modelling scenarios, but I do not think that certain scenarios can be as directly linked to changes in emissions. I suggest to modify the wording or put the scenarios in the context of studies that investigated the climate's reaction to an, unfortunately highly unrealistic, stabilization or drop in atmospheric greenhouse-gas concentrations. Sometimes Jostedalsbreen is mentioned as the "largest glacier on mainland Europe", sometimes as the "largest glacier of the European continent". While the second might be correct in the strict sense of plate tectonics, it is not correct given that Iceland, with its much larger ice caps, is an integer part of Europe. For these reasons, I prefer "mainland Europe".

We thank the editor for these good suggestions. The terminology was indeed not clear. Consequently, we revised the definition of "committed mass loss" in the Introduction, removing any statements of emissions,

and in line with other studies of committed mass loss. We have gone through the manuscript and corrected statements about emissions and climate, especially in light of committed mass loss, and the Reverse experiments. We now explain that the Reverse experiments represent an instant reset of the mass balance forcing after 2100 to that of 2000-2020, but we do not specify how this mass balance situation could arise from changes in carbon emissions. Similarly, the Commit experiments do not imply any assumptions about emissions nor associated temperature response, after 2100.

Finally, we corrected the continent vs mainland issue, thanks for spotting this. We now use "mainland Europe" or "European mainland" throughout.

**Reviewer 1**

**General comments**

This manuscript presents a model study that uses the Ice-sheet and Sea-level System Model (ISSM) and temperature index model forced with seNorge_2018 gridded daily mean temperature and daily total precipitation and 4 Climate models with two scenarios to simulate recent history (from 1960) and future demise of Jostedalsbreen, until 2100 and then onwards with constant climate until 2300 to assess the committed mass loss for several scenarios. Clear description of model components and observations used for the model initialisation and a thorough analysis of the results is made. This paper presents first simulations of Jostedalsbreen as a whole and has important message about its projected evolution in the future and about the difference in mass loss of different model approaches, i.e. the difference in mass loss between global/regional assessments that use flow line models to model individual outlets of complex ice caps separately, and models of whole ice masses as presented here (lines 578-589). The paper is extensive with much information, 12 Figures and 2 tables and additional 5 figures and a table in appendix. There are a few issues regarding consistent terminology and sequence of figures listed in the specific comments that if addressed would increase the clarity of the paper.

We much appreciate these encouraging words from the reviewer. We address the comments on terminology and figures below.

**Specific comments**

The use of the terms committed mass loss, 'in the pipeline' (line 103) and response time of glaciers is confusing and could be clarified by using one consistent way of describing committed mass loss due to long response time of glaciers. This would be (at least, perhaps other places as well) on page 3, lines 88-90, 101-104, page 12 line 281, page 14 line 356 and 362, line 491, 495, page 25 line 507, line 598, page 35 line 740. The explanation on lines 100-107 is a good place to define the terminology and use it consistently throughout the paper.

Good point, see our response to comments made by the Editor. We have gone through the manuscript to ensure consistent, clear terminology. We have also gone over the definition in line 103 of the original manuscript, as suggested, and specify that "committed mass loss" and "in the pipeline" are the same thing.

The figures are not discussed in the same order as appearing, causing readers to have to jump back and forth and reduces the readability. this could be improved by adding results of Future8.5-ECC on Figure 2

and rearranging the sequence of figures to fit better the flow of the text.

Figure 2 shows both historical and future results for one scenario (Future4.5-ECC), and also includes historical and future year-to-year SMB for this simulation. We considered adding Future8.5-ECC as well to Fig 2ab and de, but in the end thought it would confuse the reader, since SMB in (c) and (e) are shown for a single simulation, and adding a second simulation in (c) and (e) would clutter the figure. On readability: we spent quite some time finding the optimal sequence of figures. If the Reviewer has concrete examples on places where the reader has to jump back and forth, we are open to address these. We did correct referencing Fig. 9b to 2b on l.391, as the Reviewer suggested below.

The uses of the terminology "X and Y, due to A and B, respectively" is not always correct, for example lines 480-481, line 601, line 722, 710-711, 722, and perhaps other places, the correct way is in line 609
Rewrote lines 480–481, 601, 710–711, and 722.

The colloquial term "from scratch" sounds strange in this context, it is in several places, including line 573, the figure caption of figure A3 and in text, suggest to use "from no ice conditions" instead.
Changed wording to "ice-free conditions" or "no ice" in a few places.

The use of ice caps – glacier unit is not consistent, in lines 13, 17 on page 1, page 3 lines 57-60, page 34 lines 688-690
We think the terminology is fine, but to avoid confusion, we now define the terms "ice cap", "glacier unit" and "outlet glacier" first in the Introduction: "Ice caps are dome-shaped ice masses with radial flow, often covering the underlying highland topography. They consist of numerous connected glacier units (e.g. Zemp and Haeberli, 2007), which also can be referred to as outlet glaciers."

**Technical comments**

Line 131 – suggest to replace "surface" catchments with "ice" – or "water" catchments
Changed to "Glacier ice-surface catchments" to be more precise.

Line 190 here can be clarified, is the velocity along surface slope that is the 2D velocity produced?
We now clarify that also the 2D-velocity (= displacement) is in "line of sight" and not "along slope". Cited from the GAMMA Software user manual (https://www.gamma-rs.ch/gamma-software/gamma-software):
"The differential interferometric phase is proportional to the displacement in the direction of the line of sight (LOS) from the radar to the target, also called the SAR look vector. [...] Differential interferometry only measures the component of the 3D deformation vector that is in the direction of the look vector. If the 3D displacement vector is assumed to be entirely vertical or horizontal, then the LOS displacement can be scaled to determine the total vertical or horizontal displacement. " We did not follow the assumption that all displacements are necessarily only horizontal in all areas of the ice cap and hence assumed LOS velocity to be most appropriate.

Line 215 Oldedalen is not shown on Figure 1, could be added?
We labelled the location of Oldedalen in Figure 1.

Line 217-218 this assumption is not discussed again, could add some discussion of the consequences of this assumption on page 33, line 688?

See reply below to comment on same line from Reviewer 2.

Line 229, here could the term cost function be added to clarify: The misfit between the modelled u and observed ice velocities, the cost function J(u, alpha)

Done

Line 258 is the RMSE difference between the modelled and observed velocities?

Yes. Clarified.

Line 261 what does "more or less accurate" mean? Can it be quantified?

We agree that this could be elaborated further and now write: "We assume that the ice cap was in steady-state in the 1960s. This assumption may be most accurate for steep and short glaciers with fast response times, which mean they are more likely to be in tune with the ambient climate (e.g. Jóhannesson et al., 1989). In contrast, larger outlet glaciers may still adjust to a long-term climate signal and hence not be in equilibrium with 1960 conditions.

Line 264 is the RMSE difference between the observed and modelled thickness? (rewrite for clarity)

Clarified.

Line 269 suggest to replace "dynamics" with "parameters"

Done

Line 270 how many? Suggest to replace "positions" with "length" or add terminus positions

Added "terminus" to clarify. Several of the glaciers with front variation observations (NVE, 2025), have small changes in frontal positions from around 1960–1989. These are Stigaholtbreen, Austerdalsbreen, Bødalsbreen, Nigardsbreen, Brenndalsbreen, Tuftebreen, and to some degree Briksdalsbreen (stable 1960s and 1970s, although advance in 1980s). A few glaciers displayed considerable changes, namely Lodalsbreen, Fåbergstølsbreen, and to some degree Supphellebreen (stable in the 1950s, advance in 60s–80s). Finally, several outlet glaciers have too little/no data to conclude (Kjenndalsbreen, Bergsetbreen, Tunsbergdalsbreen, Boyabreen, Vetle Supphellebreen).

Based on this we added some context to the steady-state assumption mentioned in the text.

Table 2 could references for the climate models be added to table?

Added references to EURO-CORDEX simulations (Jacob et al., 2014) and downscaled projections (Wong et al., 2016) to the table caption (in the same way as done for the seNorge dataset).

Line287 How is this found? Can be rewritten for clarification

We performed some two-way coupled historical simulations at an early stage, but the results were nearly identical to the one-way coupled historical simulations. We therefore continued with the latter for efficiency. We now mention this in the manuscript.

Line 301-302 here text can be clarified, how low RMSE?
Quantified.

Line 333 what are the monthly linear trend models based on, text can be edited for clarification
We specified that we refer to monthly linear trends in daily temperature and precipitation over the reference period.

Line 381 – what does "well within uncertainty" mean?
Clarified.

Line 385 suggest to replace "variations" with "periods" (and rewrite as in figure caption)
Changed to "...can be divided into three distinct periods".

Line 387 replace Fig. 2c) with 2 a)
Changed to "Fig. 2ac", as both SMB and ice-volume evolution is mentioned in the text.

Line 391 repalce fige 9b with 2 b)
Done.

Line 392 suggest to replace "apparent" with "modelled"
Done.

Line 395 should the be "not" in front of glacierised?
The original wording is correct. Minor clarification in text.

Line 398 strange wording of "should be considered satisfactory" suggest to rewrite for clarification
Changed to "...suggests that the model performs well in representing both ice volume and area."

Line 401 and 406 inconsistency of 30-80 m and 20-50 m
Here we refer to the typical uncertainty in ice thickness for two different glaciers, Tunsbergdalsbreen (30–80 m) and Nigardsbreen (20–50 m). We have now specified this for clarity.

Line 409 suggest to add "modelled" in front of frontal position and "longer" instead of down valley
We added "modelled" in front of frontal position. Since we are referring to frontal position and not glacier length, we do not believe it makes sense to change the wording to "longer". However, we changed the formulation to "further downvalley" for clarity.

Line 465 suggest to add "modelled" for clarification
Changed wording to "Modelled contemporary ice thickness".

Line 507 "on their way to steady state" is not clear, see comment above on response time, how long time does it take to reach steady state?

Clarified. This would take another 100 years, and is now mentioned in the text.

Line 515-516 , suggest to swap order, first state reality and then response?
Done.

line 530 suggest to replace "pathway" with "scenario" for consistency
Done.

Line 624, the sentence is strange, how is model performance dependent of observations? Is it dependent on the parameter selection?
Good point. It's rather the agreement between model and reality that depends on the uncertainty of observations (and the other factors listed). In other words, if the model was perfect, the agreement may still be off if the observations are off. This is now clarified.

Line 630 what is meant by "in theory" here? Are the model output with this high uncertainty?
This was confusing wording. We have changed to:
"Despite the unprecedented level of detail in this dataset, the uncertainties in areas without thickness observations are estimated to be up to 50–100 m, but in practice likely smaller, particularly close to radar survey lines (Gillespie et al., 2024)."

Line 637 more concise text would be useful here, are they located, or not? Are some? By how much? Or rewrite for clarity
We now quantify the disagreement between modelled and observed terminus positions for these glaciers, and elaborate on the effects seen on modelled future outlet glacier behaviour.

Line 649, suggest to replace "heavily influenced" by "controlled"
Changed to "strongly controlled" (dynamics also play a role, but SMB is the strong control).

Line 651 suggest to replace "small scale" by "spatially variable" I don't think wind redistribution or avalances are small scale, they may have small impact on the total, suggest rewriting for clarity
Done.

Line 653 suggest to replace "corrected" with "variable"
Done.

Line 663 suggest to rewrite, replae "reality" with "observations" and the next sentence implies that velocity data is not "correct" does it have high uncertainty?
Replaced "reality" with "observations" as suggested. Changed the wording about the velocity data to
"The velocity data appears to be of good quality for the large outlet glaciers (Sect. 3.2), and more uncertain for small and narrow glaciers. Comparison against the global velocity dataset suggests that several model glaciers are less dynamic compared to observations (Fig. 3d). Given accurate velocity data, this could partly explain..."

Line 671 what does "representative" mean here? - low uncertainty?
Changed from "most representative to "highest".

Line 679-680 suggest to edit, strange wording "thicknesses diverge"? simulate observations?
Changed to "...we expect and accept that some mismatches between the modelled and observation-based thickness..."

Line 696 and 697 edit (Fig A4ef to Fig A4 e) and f)
Done.

Line 713 add "high" after unrealistic?
Changed to "unrealistically high".

Line 719 suggest to add "current climate" to sentence and rewrite for clarity
Done.

Line 725 suggest to replace "reversed" with "replaced by"
Changed "reversed" to "instantly reset", as to improve clarity.

728-730 suggest to edit, this is not clear "difficult to regrow" what does that mean? "not enough" is not clear
This was indeed a bit sloppy writing. Changed to
"Jostedalsbreen would likely not regrow to a large extent if it disappeared. The current climate of the 2000s is not favourable enough to regenerate the contemporary ice cap."

Line 732 take s off appear for plural
Done.

**Missing references in**
Line 31
"This setting ranks among the most challenging real-world cases outside the polar ice sheets." We removed this sentence, as the former sentence already describes that the complex topography/glacier geometry and great variability of glacier units represents a challenging model case.

Line 117 both for temperature-index model and Bayesian approach
Added.

Line 164 (QGIS)
Added.

Line 168 for1966 DTM for entire ice cap
Added.

Line 174 ArcGIS Pro and ANUDEM should be referenced
Added.

Line 261 are there references to show the validity of the assumption?
Yes, these cross-refs point to where model choices, assumptions and input data are detailed (Sect. 3 and 4), and discussion of their limitations (Sect. 7.3). This is now clarified.

Line 278 what are the references for the reanalysis?
Added.

Line 543 reference for uplift?
Added.

Line 545, regardless of what? Suggest to rewrite for clarity
Rewritten to enhance clarity: "The large projected increases in precipitation and differences between models and scenarios highlight the considerable uncertainty in future temperature and precipitation in mountainous regions and the sensitivity of glacier projections to climate model input."

Line 560 add transGeo or Area on the SMB
Added.

Line 630 add reference to the ice thickness model here?
We already refer to Section 3.1 here, and have now also added "see Gillespie et al. 2024 for details"

Line 739 add reference for machine learning
Added.

Figure 3 the color scheme for the velocity figure is not helpful to show the variability in magnitude of velocity, could a log scale be used, or longer scale to show the lower velocities better, the map is almost completely green and does not show much.
Great suggestion, we switched to a log scale for the velocity magnitudes, with a colormap commonly used for ice velocities in other studies.

Figure 7, suggest to add Bedrock to beginning of figure caption – and explain shy the elevation is fluctuating, there appears to be some up and dow in figures c) and d) blue colors between the yellow.
Added "Elevation of bedrock..." in the caption. We are not sure here, but we think the reviewer refers to the decadal fluctuations in modelled SMB here, where the changes are not monotonously changing from blue to yellow (more negative). In contrast, modelled glacier surface elevations show a consistent lowering over time (change from blues to yellows). We have added this clarification in the caption.

Figure 8 would it be possible to show the sizes of each region by sizes of the circles? So that at glance of the figure readers would have impression of how different sizes region the different circles represent?
Thanks, good idea. We have rearranged the discs, and now use present-day ice volume to scale the disc area

for Jostedalsbreen, North, Central, and South relative to each other. Similarly, we now scale the disc area for the individual glaciers too. We have also present-day ice volume in text next to each disc, to guide the reader further.

Fiugre A2 is not referred to in text and no discussion of 10 or 20 m threshold is in the text, either add that, or delete figure
There is discussion and a figure reference on this in lines 393–398 of the original manuscript. No changes made.

**Reviewer 2**

The manuscript by Åkesson et al. presents model simulations of Jostedalsbreen ice cap in a past, present and future climate. The manuscript is very detailed and well-written, the methods are sound, the structure is clear, and the results are novel, relevant and significant. I only have minor suggestions for improvements and clarification, which are detailed below.
We are happy to read these positive comments about the manuscript. We address the specific and technical comments below.

**Specific comments**

L7: "12-74%". It could be useful to include a median / mid-range estimate here.
Added the mid-range estimate based on ECEARTH/CCLM, as was mentioned in the Conclusion.

L12: "are irreversible". Please define what is meant with irreversible here, e.g. by referring to hysteresis theory or committed mass loss.
We now explain explicitly what we mean:
"... irreversible; the ice cap would not recover to its contemporary volume if the future surface mass balance was reversed to that of the present-day."

L31-32: "The wide range of geometries ... of the world.". This is rather what you find in many ice caps (rather than glaciers) globally, with slow-flowing interiors and fast-flowing outlet glaciers.
Good point, the great diversity is indeed representative of other ice caps/ice fields. What we meant was that if one looks at individual glacier units, these vary in shape, size and dynamics, and thus have similar characteristics to many mountain glaciers around world. This is now clarified:
"The wide range of characteristics of the individual glacier units found at Jostedalsbreen make them representative of glaciers found in many glacierised regions of the world."

L45-49: In addition to highlighting the need for 3-D higher-order modelling, it could be emphasized that shallowness assumptions in commonly used ice flow models are not justified for complex ice cap systems.
Thanks for this suggestion. This was already in mind in l.43–48, but it doesn't hurt to be more explicit. We have added:
For example, assumptions of shallowness (Hutter, 1983) commonly used in many ice-flow models are not

justified for complex ice caps.

L50-67: To my taste, the discussion of global glacier modelling is a bit too lengthy and could be condensed to a few sentences.
We moved the content on the mass balance-elevation feedback two paragraphs up, since this is strictly not something only relevant for global models. Moreover, we condensed the paragraph on global glacier modelling from 18 to 10 lines.

L84-85: "Some of these models struggle ... because each glacier is treated individually". It could be good to include a reference to which model(s) that applies (e.g. Millan et al. 2022).
Added relevant references: Farinotti et al 2019 and Millan et al 2022

L218: "the lake surface is considered the bedrock topography". Does this create a wall that slows down motion (and induces thickening) near the fronts?
In principle yes, although it would be rather a "bed step" at the front than a wall, to be precise. This applies to the proglacial lakes in front of the outlet glaciers Austdalsbreen and Sygneskarsbreen in the north. These lakes are dammed for hydropower purposes. The bathymetries are not well known (Andreassen et al., 2020), but the calving front of Austdalsbreen is currently grounded at 1175 m a.s.l in the centre and 1200 m a.s.l at the margins; that means 1200 m flat "lake terrain", as used in the simulations, would introduce a ridge, but not a wall that essentially stops the glacier. Treating the lake surface as bedrock would mainly influence potential glacier advance, but this does not generally not occur in our simulations. Neglecting calving as a mass loss component could of course lead to underestimate of mass loss for these specific glaciers.
We have modified the text to the following:
"Two of the 81 glacier units are lake-terminating glaciers (Austdalsbreen and Sygneskarsbreen in the north), which account for about 3% of the total ice-cap volume. Frontal ablation is not included in the model, and the lake surface is considered the bedrock topography. Neglecting iceberg calving and subaqueous melt may lead to underestimation of glacier mass loss so long these glaciers remain lake-terminating. The uncertainty in bedrock elevation may affect potential glacier advances, however glacier retreat dominates in our simulations."

L228: "We constrain the local values of the basal friction parameter". I suppose the bed topography is kept fixed following Gillespie et al. (2024). Does this imply that all errors in both ice flow model physics, mass balance forcing and bed topography are compensated for by basal friction adjustments? It could be worth adding some discussion on this in the Discussion section.
This is a good point, we have added some discussion of this at the end of Section 7.3:
"The spatially variable friction field obtained aims to represent the bulk effect of all interactions between ice, water, rock and sediments at the ice-bed interface. Still, friction parameter adjustments are essentially also compensating for potentially missing physics and uncertainties in the bed topography, ice-flow physics and surface mass balance forcing."

L237: "For the rest of the ice cap". What would happen if you apply the above approach (the adjoint-based minimization) also to "the rest of the ice cap"? Would the results be clearly worse than with the elevation-dependent function?
Thanks for bringing this up, we have tested this. The results for the historical simulation using a basal

friction inversion applied everywhere are not too bad, but still worse than with the hybrid approach used in the manuscript, where the inversion is done for the two largest outlets Nigardsbreen and Tunsbergsdalsbreen, and an elevation-dependent friction parameter is used elsewhere. At the end of the historical simulation, the RMSE for the velocity misfit using the inversion everywhere is 22 m/a, actually 4.4% lower than the RMSE of 23 m/a using the hybrid friction approach. For thickness, the end-of-historical RMSE is 27 m using the inversion everywhere, 12.5% higher misfit than a RMSE of 24 m with the hybrid approach. The observed 2020 ice volume is $70.6 \pm 10.2$ km$^3$. The modelled 2020 ice volume with a friction inversion everywhere is 68.1 km$^3$ (3.5% mismatch), while with the hybrid approach used in the manuscript the modelled volume is 70.3 km$^3$ (0.4% mismatch), see Figure 1 below. The slightly lower volume using the friction inversion everywhere stems from a lower ice volume (thinner ice) at end-of-spinup (start of historical simulation), as well as slightly differing temporal evolution (Figure 1 below). Based on this we feel confident in keeping the hybrid approach as outlined in the original manuscript. We added a short note towards the end of Section 7.3: "For example, using a friction inversion for the entire ice cap increases the mismatch with present-day observed ice volume from 0.4% using the current hybrid approach to 3.5% using the inversion everywhere."

Section 4.3: It could be useful to briefly mention how the SMB model distinguishes precipitation falling as rain or snow. This is relevant for understanding the future SMB description.
We added a brief explanation of this by amending the sentence on L243: "To simulate SMB, we use the model of Sjursen et al. (2023, 2025) where melt is modelled using a temperature-index approach and accumulation is the sum of solid precipitation, assuming a linear transition between solid and liquid precipitation around a temperature threshold."

L256: "obtain the best possible representation of ice velocities". Aren't velocities for Tunsbergsdalsbreen and Nigardsbreen forced to match observations by minimizing the cost function (eq. 2)?
Yes, but even with the inversion done using the cost function for Tunsbergsdalsbreen and Nigardsbreen, there are mismatches between observed and modelled velocities for these glaciers too. We changed "best possible representation" to "an optimal representation".

L260: "we assume that the ice cap was in steady-state in 1960s.". Maybe this sentence could be moved to after L270 when the spin-up process is described. Then it is directly clear how the steady-state is achieved.
Moved as suggested.

L264-268: It could be good to merge/move the part about friction inversion with/to section 4.2 for clarity.
We considered this, but the optimisation of friction parameters is part of the calibration and spin-up process, and we therefore feel it belongs to the current Section 5.1. No change in response to this comment.

L269-270: "Once the optimal model dynamics is obtained, we perform a 250-year spinup...". Have you considered/tested alternative methods to spin up the model in 1960? The current method advantageously can use the detailed known bed as a boundary condition, but drawbacks are the steady state assumption in 1960 and lack of agreement between modelled and observed surface height. I wonder whether the authors considered using an inverse approach as in Frank and Van Pelt (2024), which advantageously would have been faster and would have removed the steady-state assumption. On the downside, it would have implied the need not to fix the bed. Please consider adding some discussion on this.

[Figure]

Figure 1: Comparison of historical ice-cap evolution using a friction inversion applied to the entire ice cap (left panels) and a hybrid friction approach (right panels), as shown in Figure 2 in the manuscript.

Indeed, we tested other approaches to obtain the initial historical geometry, including considerable efforts without a steady-state assumption. To this end, we started from the 1966 surface DEM and ran a relaxation for only a few years. This approach resulted in significant model drift of the ice-surface topography in the historical simulation. During this process, we tried several approaches/parameters to adjust the friction parameters (mentioned in Sect. 7.3), as well as several different SMB forcings (mean 1960s, mean 1960s and 1970s, mean 1960s to 1980s, various manual adjstuments of the SMB field, etc). We therefore decided to switch to the more practical steady-state assumption, as this ensured no model drift.

We added the following to Section 7.3 discussion:

"We also tested to obtain the initial historical geometry without a steady-state assumption, starting from the 1966 surface DEM and running a relaxation for only a few years. While surface data artifacts dissipated, this approach resulted in unrealistic model drift of the ice-surface topography during the historical simulation. We therefore switched to the practical steady-state assumption."

An inverse approach with historical ice thickness (bed topography) as a "free parameter" (Frank and van Pelt, 2024) is an interesting idea. Gillespie et al. (2024) compared the thickness data for Jostedalsbreen with that inferred by Frank and van Pelt (2024). The latter show a tendency to overestimate thickness on outlet glacier tongues but in general show an ice-thickness distribution that is very consistent with the thickness dataset from Gillespie et al. (2024). We have added:

"An inverse approach to find the ice thickness (Frank and van Pelt, 2024) could also have been considered, yet here we take advantage of the detailed newly collected thickness dataset from Gillespie et al. (2024)."

L357-358: "using the mean annual SMB of 2001-2020". Since the present-day SMB is likely more negative than the 2001-2020 average, it is more a "Commit-2010" scenario that is tested here. I.e. we are already 15 years too late for this scenario ;)

Glaciological mass balance measurements on Nigardsbreen and Austdalsbreen, as well as our simulated SMB, actually show a more negative SMB in the early 2000s, i.e. more negative in the 2001-2010 period than the 2011-2020 period (likely influenced by variations in both summer temperature and winter precipitation. We therefore believe that the 2001–2020 average SMB is representative of the present day climate. No changes were made.

L361: "The Commit experiments". I am not fully convinced whether the Commit4.5 and 8.5 results are particularly useful. It is rather arbitrary to use the end of the 21st century as the start of a stable future climate. It also adds two more runs to an already large suite of experiments.

We think that the Commit experiments is an informative way to assess the long-term response of the ice cap to climate change. We believe this is understudied, as we emphasise in the Intro, l.101-102 of the original manuscript: "To understand the full implications of current and near-future climate change, we however also need to consider future inertia and longer timescales." To be clear, the end-of-the-century climate forcing in these experiments is not the SMB in a single year (year 2100), but the mean SMB of the two final decades (2081–2100). One could of course have used a mean over three decades, or only one, but since a 20-year time period is used for the present-day climate (SMB 2000–2020), we felt this was appropriate. We added a short note on this in Discussion Section 7.3 on SMB:

"In our Commit-experiments, the mean SMB 2081–2100 is used as a constant climate forcing. Other time periods could be considered to represent the end-of-the-century climate, but the mean over two decades was chosen in order to be consistent with the two-decade mean for present-day SMB in the CommitNow experiment."

L373-377: "The historical simulations ... future projections.". This is more discussion or introduction than results.
Yes, indeed. This paragraph was meant as an introduction to the results, a help for the reader to contextualise the Results subsections. We have edited this paragraph slightly.

L379: "changed very little since the 1960s". How much of this may be due to the steady state assumption?
Removed "very". Observed ice-volume reduction from 1966 to 2020 is 4 km$^3$, while the modelled change is almost zero (Fig. 2a). In this light, the 4 km$^3$ may be a result of the steady-state assumption, resulting in a too small ice cap in the 1960s. On the other hand, the uncertainty in observed ice volume for both the 1960s and present-day is around 10 km$^3$, so it's not a straightforward comparison. We now mention this after line 380 in the original manuscript.

L379-387: A general comment, there are quite many short paragraphs in the manuscript. Please consider merging them with neighbouring paragraphs where possible.
We realise that this is partly a matter of writing style and personal preference. We have now merged the first three paragraphs in Section 6.1. We also merged paragraphs with previous ones at lines 114, 468 and 496.

L397-398: "This sensitivity analysis ... satisfactory.". I wonder why the model forms these large thin ice areas. Is it an artefact of the description of the frontal and lateral boundary conditions and/or flow speeds near the glacier boundaries?
We thought about this too. We are not sure whether the frontal or lateral boundary conditions may have an affect, as ice is allowed to move freely. It could be a result of inappropriate friction values in some areas, resulting in over- or underestimated flow resistance; the latter is more likely to cause large, thin ice areas. It could be an artefact of the mesh size, where some large mesh elements gets covered by ice during periods of positive SMB, and then contribute disproportionally to glacier area. Also, many of these areas probably should not be considered glacier ice, but rather ice patches - if ice is very thin it does not flow much/at all. Meanwhile, there are some of the thin-ice areas that are connected to the main ice cap, which should probably be considered as glacier ice. The definition of glacier ice is that it is perennial, and that it flows. Maybe this could be used as a double criterion when talking about modelled glacier area. This discussion of the 10 m vs 20 m thickness-threshold analysis in the manuscripts illustrates that it is not straightforward to determine what to consider model area. This all implies that comparisons between observed and model area, in this case may be misleading.

L399: "Overall, the modelled thickness agrees well ...". This is a good result! Since both velocity (to some extent) and bed topography are constrained with observations, the main errors in ice thickness are errors in the ice flux being too small or large which in turn results from errors in the surface mass balance. Would you agree that most of these thickness discrepancies are due to mass balance uncertainty?
Thank you! We can see the reasoning, but there is some more nuance here. Even though velocity and bed topography are constrained by observations, the observations have errors/uncertainties, as discussed in several places in the manuscript. For example, the velocity observations are not of good quality everywhere, especially in slow-flowing regions, as mentioned in the text. These velocity data also provide limited temporal information (variability). This means that friction coefficients are uncertain over the modelled historical timespan, let alone in future simulations. We added a short note on this in Section 7.3. Meanwhile, the modelled surface mass balance is also constrained by observations (see the dedicated paper on Jostedalsbreen modelled SMB by Sjursen et al 2024 JGlac, referenced in the manuscript). We agree that the main errors in modelled ice thickness are errors in the ice flux, but find it difficult to make general statements of which dataset/factor that contributes most. It may be possible to deduce this on the individual glacier level. If we for example look at the well-studied Nigardsbreen, a glacier with well-observed ice thickness and reasonable velocity data, it is tempting to conclude that most of the thickness discrepancies are due to SMB uncertainty. Even so, the impact of inaccurate model dynamics (friction) for a fast-flowing glacier like Nigardsbreen will be much bigger than for a slow-flowing glacier unit.

L428: "gains in future precipitation". Would not much of that fall as rain instead of snow?
The gains in winter precipitation for this scenario are extremely rapid (i.e. large increase already over the 2021-2040 period) such that, although winter temperature also increases somewhat, the results indicate that much of this precipitation will still fall as snow in the high-lying accumulation area of Jostedalsbreen. We elaborated on this in the text and added a reference to Fig. A5: "..gains in future winter precipitation, along with mainly negative winter temperature and little change in summer temperature (Fig. A5).

L439: "Considering the mid-range climate forcing." It could have been considered to focus the future analysis only on the mid-range scenarios. The current spread in the climate projections seems unnecessarily large, especially for RCP 8.5. It is also odd that the average of RCP8.5 mass loss is lower than for RCP4.5. We see the point, and indeed do focus most of the analysis and figures on based on the mid-range climate forcing (ECEARTH/CCLM). We also discuss (discard) some of the more extreme (unrealistic) climate projections in the text. We considered to not include all projections in the figures, to avoid that the reader gets the impression that future glacier evolution can go in (almost) any direction, and that the uncertainties appear higher than they are in practice. In the end we still wanted to include all projections in figures for completeness, and to hide any information. Showing the complete spread also highlights that climate projections in mountain regions diverge greatly; dare we say that this is the greatest source of uncertainty in glacier projections? This calls for further research in this direction, which is mentioned in the manuscript (Sect 7.1, and Conclusion).

Figure 8: It is hard to see which circle diagram belongs to which glacier on the map. Furthermore, I would prefer discs indicating how much mass is left rather than how much is lost. That would be more intuitive. I.e. a full circle means all ice remains.
We have remade this figure, see response to Reviewer 1 above. We now show different colours and line styles for the outlines and discs representing the different regions and glaciers.
Showing ice remaining vs ice lost in the discs we think is a matter of personal preference. We we wanted to focus on the question "how much ice will be lost", and thus framed both the text and Figure 8 based on this.

Section 6.2.4 (L500-510): The hysteresis effect could be discussed here. Thick (flat) ice caps can maintain a positive or zero SMB because of the high surface elevation. When the average elevations drop too much, the mass loss becomes 'irreversible' unless a very cold period happens over an extended period of time (i.e. even much colder than the 20th century climate). This hysteris effect applies mostly to low-sloping glaciers,

ice caps and e.g. the Greenland Ice Sheet.

Great point. We added some discussion of this at the end of Section 6.2.4

Section 7.2: Just an idea, there is currently a lot of information and relative change numbers for different glaciers in this section. It could be helpful for the reader to summarize the results in a table showing relative changes for Jostedalsbreen (this study) and other glaciers at different points in the future.

We have added a table of future projected changes for glaciers in Norway from our and previous study.

L678-681: "Summary. ...". This could be removed or moved to the conclusions.

Removed.

Section 7.4: There is quite a bit of repetition here of the introduction, e.g. there is no need to again refer to global glacier retreat in a warming climate or impacts on society. Please consider shortening the text.

We removed the sentence on impacts on society, as this indeed was already mentioned in the Introduction. The rest of Section 7.4 talks about future mass loss in different regions, highlighting previous projections on ice caps and ice fields and puts Jostedalsbreen into this context. The Introduction does not really contain these aspects; we therefore feel that this information is appropriate in Section 7.4, and kept it as is.

**Technical corrections**

L6: "3-d" –¿ "three-dimensional (3-D)".

Changed to "three-dimensional" in the abstract and to "three-dimensional (3-D)" and "one-dimensional (1-D)" at first mention in the main text. Changed "3-d" and "1-d" to "3-D" and "1-D", respectively, throughout the text.

L42: "Meur" –¿ "Le Meur".

Done.

L84: "Pelt" –¿ "Van Pelt".

Done.

L174: "ANUDEM". Please define the acronym.

Done.

L191: "GAMMA Remote Sensing software". Please include a reference.

Done.

L206: "is" –¿ "are.

Done.

L364-365: "To this end, ... is used as a forcing". It could be added that this applies to after 2100.

Added "..from 2101 until 2300.

Figure 3 caption: Please remove brackets around 2022.

Done.

**References**

Andreassen, L. M., Elvehøy, H., Jackson, M., and Kjøllmoen, B.: Glaciological investigations in Norway 2019, NVE Rapport 34-2020, Tech. Rep. 34-2020, Norwegian Water and Energy Directorate (NVE), URL $https://publikasjoner.nve.no/rapport/2020/rapport2020_34.pdf$, 2020.

Frank, T. and van Pelt, W. J. J.: Ice volume and thickness of all Scandinavian glaciers and ice caps, Journal of Glaciology, pp. 1–14, https://doi.org/10.1017/jog.2024.25, 2024.

Gillespie, M. K., Andreassen, L. M., Huss, M., de Villiers, S., Sjursen, K. H., Aasen, J., Bakke, J., Cederstrøm, J. M., Elvehøy, H., Kjøllmoen, B., Loe, E., Meland, M., Melvold, K., Nerhus, S. D., Røthe, T. O., Støren, E. W. N., Øst, K., and Yde, J. C.: Ice thickness and bed topography of Jostedalsbreen ice cap, Norway, Earth System Science Data, 16, 5799–5825, https://doi.org/10.5194/essd-16-5799-2024, publisher: Copernicus GmbH, 2024.

Hutter, K.: Theoretical glaciology: material science of ice and the mechanics of glaciers and ice sheets, vol. 1, Springer, https://doi.org/10.1007/978-94-015-1167-4, 1983.

Jacob, D., Petersen, J., Eggert, B., Alias, A., Christensen, O., Bouwer, L., Braun, A., Colette, A., Déqué, M., Georgievski, G., Georgopoulou, E., Gobiet, A., Menut, L., Nikulin, G., Haensler, A., Hempelmann, N., Jones, C., Keuler, K., Kovats, S., and Yiou, P.: EURO-CORDEX: New high-resolution climate change projections for European impact research, Regional Environmental Change, 14, https://doi.org/10.1007/s10113-013-0499-2, 2014.

Jóhannesson, T., Raymond, C., and Waddington, E.: Time–Scale for Adjustment of Glaciers to Changes in Mass Balance, Journal of Glaciology, 35, 355–369, https://doi.org/10.3189/S002214300000928X, 1989.

NVE: Climate indicator products, `https://glacier.nve.no/glacier/viewer/ci/en/`, Last accessed on 28 Aug 2025, 2025.

Sjursen, K. H., Dunse, T., Tambue, A., Schuler, T. V., and Andreassen, L. M.: Bayesian parameter estimation in glacier mass-balance modelling using observations with distinct temporal resolutions and uncertainties, Journal of Glaciology, p. 1–20, https://doi.org/10.1017/jog.2023.62, 2023.

Sjursen, K. H., Dunse, T., Schuler, T. V., Andreassen, L. M., and Åkesson, H.: Spatiotemporal mass balance variability of Jostedalsbreen Ice Cap, Norway, revealed by a temperature-index model using Bayesian inference, Annals of Glaciology, 66, 1–18, https://doi.org/10.1017/aog.2024.41, 2025.

Wong, W. K., Haddeland, I., Lawrence, D., and Beldring, S.: Gridded 1 x 1 km climate and hydrological projections for Norway, NVE report 59/2016, Norwegian Water Resources and Energy Directorate (NVE), Oslo, Norway, 2016.

Zemp, M. and Haeberli, W.: Glaciers and ice caps. Part I: Global overview and outlook. Part II: Glacier changes around the world, UNEP, Nairobi, URL `https://doi.org/10.5167/uzh-40427`, 2007.

---

## Author Response (AR2)

**Dear Editor,**

We thank the Editor for taking the time for a final review of the manuscript, and accepting the manuscript in TC. We have made the following final corrections, in response to the Editor's comments.

Line 23: I think the definition of 'outlet glaciers' is not accurate, it could be understood in a way that any glacier catchment that is part of an ice cap is an outlet glacier. I suggest replacing "which also can be referred to as outlet glaciers." with something like "which are referred to as outlet glaciers where their ablation areas take the shape of glacier tongues." Changed to "...which are referred to as outlet glaciers if their lower reaches are separated by mountain areas."

Line 24: I suggest replacing "input" with "mass loss", I think this would be clearer. Done

Lines 230/231: Should the Andreassen citation stand in brackets instead of between semi-colons?

Done

Line 237: Maybe replace "so long" with "as long as" or "while". Changed to "as long as"

Line 305: "... has to little data to conclude." I think it should read "have" (plural) and unclear to me what to conclude. Maybe replace with something like "have too little data to determine how close they were to a steady-state".

Done

Lines 539-540: "half of the ice cap would still disappear." The statement is somewhat unspecific, this could be stated more precisely, indicating e.g. the percentage loss in surface area and ice volume.

Changed to "almost half (47\%) of the ice-cap volume would be lost"

Figure 7, caption: Is the full stop after "transects on Jostedalsbreen" in the right place? - Done

Line 688: two times "and" in a row. Is this wanted?

Removed

**Code and data availability:** This section does not mention the DTMs that were constructed for 1966 and 2020. Optimally, for reproducibility of the study they should also be made publicly available. I assume the Gillespie et al. (2024) dataset is already openly accessible, but is it planned to make the DTMs also available upon publication? Or are there restrictions due to copyrights that do not allow making them open source?

Yes, we will make the 1966 DTM available. We also added a few further details here about the access to model code and data.

**Colour schemes in figures for readers**

We have checked the proposed figures using the proposed Coblis – Color Blindness Simulator (<a href="https://www.color-blindness.com/coblis-color-blindness-simulator/">https://www.color-blindness.com/coblis-color-blindness-simulator/</a>). We found that:

- Fig. 4: We think this looks good in the Color Blindness Simulator, except for the rare type of color blindness called Monochromacy/Achromatopsia. Since this occurs with a frequency of only 1:30 000 to 1:50 000 of people, we think the figure is fine as is.
- Fig 9. Same as Fig. 4.
- Fig. 11. Same as Fig. 4.
- Fig. A5. Looks good for all types of color blindness in the simulator.

**Highlight article and short summary**

Thank you for picking our paper as "Highlight article". We have not made any changes to the 500-character short summary from the original submission.

Best wishes,

the authors